# LANGUAGE MODEL CASCADES:
# TOKEN-LEVEL UNCERTAINTY AND BEYOND

**Neha Gupta, Harikrishna Narasimhan, Wittawat Jitkrittum, Ankit Singh Rawat,**
**Aditya Krishna Menon, Sanjiv Kumar**
Google Research, New York
{nehagup, hnarasimhan, wittawat, ankitsrawat, adityakmenon, sanjivk}@google.com

## ABSTRACT

Recent advances in language models (LMs) have led to significant improvements
in quality on complex NLP tasks, but at the expense of increased inference costs.
*Cascading* offers a simple strategy to achieve more favorable cost-quality trade-
offs: here, a small model is invoked for most "easy" instances, while a few "hard"
instances are deferred to the large model. While the principles underpinning cas-
cading are well-studied for classification tasks — with deferral based on predicted
class uncertainty favored theoretically and practically — a similar understanding
is lacking for generative LM tasks. In this work, we initiate a systematic study of
deferral rules for LM cascades. We begin by examining the natural extension of
predicted class uncertainty to generative LM tasks, namely, the predicted *sequence*
uncertainty. We show that this measure suffers from the *length bias* problem, ei-
ther over- or under-emphasizing outputs based on their lengths. This is because
LMs produce a *sequence* of uncertainty values, one for each output token; and
moreover, the number of output tokens is variable across examples. To mitigate
this issue, we propose to exploit the richer token-level uncertainty information im-
plicit in generative LMs. We argue that naïve predicted sequence uncertainty cor-
responds to a simple aggregation of these uncertainties. By contrast, we show that
incorporating token-level uncertainty through learned *post-hoc deferral rules* can
significantly outperform such simple aggregation strategies, via experiments on a
range of natural language benchmarks with FLAN-T5 models. We further show
that incorporating embeddings from the smaller model and intermediate layers of
the larger model can give an additional boost in the overall cost-quality tradeoff.

## 1 INTRODUCTION

Recent advances in generative language modeling have yielded a series of Transformer-based mod-
els with remarkably improved *quality* on complex NLP tasks (Radford et al., 2018; Raffel et al.,
2020; Brown et al., 2020; Black et al., 2022; Hoffmann et al., 2022; Chowdhery et al., 2022; Wei
et al., 2022; Chung et al., 2022; Tay et al., 2023; Anil et al., 2023; Touvron et al., 2023; Team
et al., 2023). Unfortunately, such models also involve significantly increased *inference costs*, which
has motivated a series of efforts at reducing the same. These span careful infrastructure optimiza-
tion (Chowdhery et al., 2022; Pope et al., 2022; Sheng et al., 2023), rethinking the autoregressive
decoding that underpin Transformers (Stern et al., 2018; Leviathan et al., 2023; Chen et al., 2023a;
Sun et al., 2023), modifications of the underlying model architecture (Dao et al., 2022), and model
compression strategies (Frantar & Alistarh, 2023; Agarwal et al., 2023).

*Cascading* is one simple strategy to achieve more favorable cost-quality tradeoffs via *adaptive* in-
ference. In a two-model cascade, a small model is invoked for most "easy" instances, while a few
"hard" instances are deferred to a large model. Cascades have been widely explored in the vision
domain (Viola & Jones, 2001; Trapeznikov & Saligrama, 2013; Bolukbasi et al., 2017; Huang et al.,
2018; Rawat et al., 2021; Kag et al., 2023; Jitkrittum et al., 2023), and have seen increasing adop-
tion within NLP (Mamou et al., 2022; Varshney & Baral, 2022; Khalili et al., 2022; Dohan et al.,
2022; Chen et al., 2023b;a). Importantly, cascades can be implemented in a black-box fashion over
existing models, and do not necessitate any additional training.

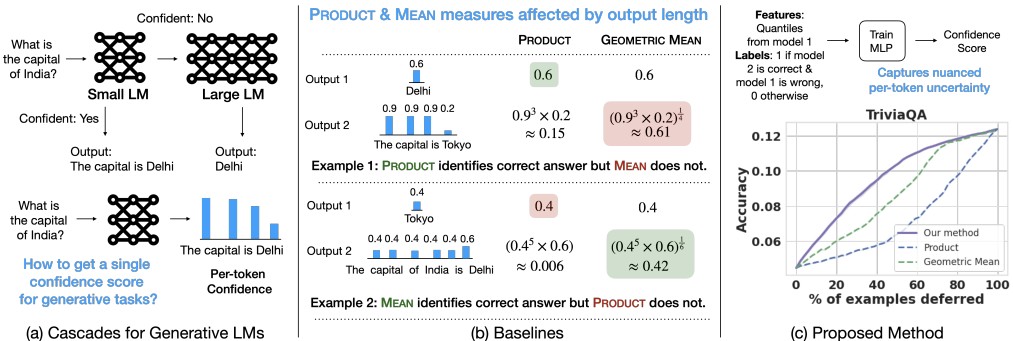

Figure 1: **(a)** In cascades, small models are used for easy instances whereas hard instances are routed to larger models. For generative LMs, the key challenge is to design a deferral rule based on uncertainties from multiple tokens. **(b)** Standard baselines which take the product and geometric mean of the probabilities are affected by the length of the output and perform sub-optimally. **(c)** Our proposed solution captures nuanced per-token uncertainty and outperforms both baselines.

The key challenge in cascading is to design a *deferral rule* which decides whether to defer an input to the larger model. The principles underpinning optimal deferral rules are well known for the classification setting, where the standard recipe is to employ the small model's prediction *confidence*, as canonically measured by its softmax probability output (*Chow's rule* (Chow, 1970)). This simple deferral rule is remarkably hard to surpass in most natural settings (Jitkrittum et al., 2023).

However, the narrative is more complex for generative LMs. While one can naïvely translate Chow's rule for such models based on the softmax probability of the output sequence, this suffers from a *length bias* issue: one tends to defer longer predictions, regardless of quality. Further, simply normalizing the probability by sequence length tends to over-correct this bias, and defer shorter predictions. Intuitively, such naïve translations of Chow's rule ignore a key distinction between the classification and generative LM setting: the former involves a *single* probability distribution over labels, while the latter involves a *sequence* of distributions over multiple tokens of the LM output; moreover, the number of output tokens differs across examples. This variability complicates summarizing the sequence of uncertainty (or confidence) values into a single deferral score.

To mitigate the length bias and capture fine-grained information from the uncertainty vector over tokens, we propose to use *quantiles* over the vector. Via experiments on a range of NLP benchmarks and FLAN-T5 models, we show that these quantiles can capture rich and complementary sources of uncertainty information from the uncertainty sequence vector and do better than the simple aggregation schemes like sum and average. However, we observe that there is no fixed quantile value which works across all datasets. This motivates us to *learn* a deferral rule based on these quantile values as features, which can combine the strengths of these different quantile scores. We show that our trained deferral rule is the most consistently performant method compared to all the natural baseline aggregation strategies. We further show that using embeddings from the smaller model and intermediate embeddings from the larger model can give further performance improvement.

To summarize, our contributions are:

(i) We show that simple sequence-level LM confidence measures for deferral can yield strongly sub-optimal cost-quality tradeoffs, owing to a length bias issue (§3.5).

(ii) We introduce *token-level uncertainty* in the form of distribution quantiles, and show that they can yield to consistently more effective cost-quality tradeoffs, owing to their finer-grained measure of uncertainty. However, there is no *fixed* quantile which works well across all settings (§3.5).

(iii) We propose a post-hoc deferral rule trained on quantile features, and show it can outperform all other strategies on a range of NLP benchmarks for FLAN-T5 models (§4.3). We further demonstrate that using the large model's *intermediate embeddings* can significantly boost performance.

## 2 BACKGROUND AND PROBLEM SETUP

In this section, we discuss the relevant background and set up the problem of LM cascades.

**Language models (LMs).** Given a finite vocabulary $\mathcal{V}$ (e.g., tokens derived from Sentence-Piece (Kudo & Richardson, 2018)), a *language model* (*LM*) defines a distribution $p(\cdot \mid \boldsymbol{x}) \in \Delta(\mathcal{V})$ over all possible tokens given any *context* $\boldsymbol{x} = (x_1, \ldots, x_m) \in \mathcal{V}^m$. This in turn defines a distribution over *sequences* $\boldsymbol{y} = (y_1, \ldots, y_n) \in \mathcal{V}^n$ for any $n \in \mathbb{N}_+$, with $p(y_1, \ldots, y_n \mid \boldsymbol{x}) = p(y_1 \mid \boldsymbol{x}) \cdot \prod_{i=1}^{n-1} p(y_{i+1} \mid \boldsymbol{x}, y_1, \ldots, y_i)$ via the chain rule of probability.

LMs based on Transformers (Vaswani et al., 2017) have proven increasingly popular in recent years. Such models are typically *pre-trained* on large corpora based on self-supervised objectives (Radford et al., 2018; Devlin et al., 2019; Raffel et al., 2020; Brown et al., 2020; Tay et al., 2022; Anil et al., 2023). These objectives involve different (input, output) pair constructions $(\boldsymbol{x}, \boldsymbol{y})$ (e.g., masking out the next token), upon which one minimizes the *cross-entropy* or log-loss, i.e., $-\log p(\boldsymbol{y} \mid \boldsymbol{x})$.

At inference time, given a trained LM and any input context $\boldsymbol{x}$, it is common to perform either *classification* or *generation*. In the former, given a set of predefined choices $\mathcal{C} = \{\boldsymbol{c}_i\}_{i \in [L]}$ (e.g., { yes, no }), one scores each $p(\boldsymbol{c}_i \mid \boldsymbol{x})$ and returns the highest scoring choice. In the latter, one performs *sampling* from $p(\cdot \mid \boldsymbol{x})$ to produce a suitable output string response, e.g., by temperature (Ficler & Goldberg, 2017), top-$k$ (Fan et al., 2018), or nucleus sampling (Holtzman et al., 2020).

**Model cascades.** Cascades are a simple, generic strategy to improve the inference cost-quality trade-off (Wang et al., 2022). Given a collection of models of varying inference cost, the key idea is to perform *adaptive* inference: "easy" samples are afforded less computation compared to "hard" samples. Concretely, for any test input, one first executes the lowest cost model, and uses a *deferral rule* to determine whether to terminate with its prediction, or to invoke the next cheapest model. Cascades can reduce the average inference cost if only a small fraction of inputs are deferred.

Cascades have a long history of usage in vision (Viola & Jones, 2001; Huang et al., 2018; Wang et al., 2018), where they are often applied for *classification* problems. Given an instance space $\mathcal{X}$ and label space $\mathcal{Y}$, the classification problem seeks a *classifier* $h \colon \mathcal{X} \to \mathcal{Y}$ with good *average quality* under some distribution $\mathbb{P}$, as measured by $\mathbb{E}_{(x,y) \sim \mathbb{P}}[q(x, y, h(x))]$ for some $q(x, y, h(x)) \in \mathbb{R}_+$. In the simplest case, $q(x, y, \hat{y}) = \mathbf{1}(y = \hat{y})$ measures the classifier *accuracy*.

Now suppose we have two classifiers $h^{(1)}, h^{(2)}$, with inference costs (e.g., latencies) $c^{(1)} \ll c^{(2)}$. Operationally, a cascade first invokes the "small" model $h^{(1)}$, and then applies a deferral rule to decide whether to either *defer* prediction to the "large" model $h^{(2)}$, or terminate with $h^{(1)}$'s prediction. More precisely, let $r \colon \mathcal{X} \to \{0, 1\}$ denote the deferral rule, where $r(x) = 1$ denotes that we wish to defer to the large model. Then, the cascaded classifier is (Kag et al., 2023; Jitkrittum et al., 2023):

$$h^{\mathrm{cas}}(x) = \mathbf{1}(r(x) = 0) \cdot h^{(1)}(x) + \mathbf{1}(r(x) = 1) \cdot h^{(2)}(x).$$

Given an input $x$, the corresponding cascade quality and cost are:

$$Q(x, y, h^{\mathrm{cas}}(x)) \doteq \mathbf{1}(r(x) = 0) \cdot q(x, y, h^{(1)}(x)) + \mathbf{1}(r(x) = 1) \cdot q(x, y, h^{(2)}(x))$$
$$C(x, h^{\mathrm{cas}}(x)) \doteq \mathbf{1}(r(x) = 0) \cdot c^{(1)} + \mathbf{1}(r(x) = 1) \cdot (c^{(1)} + c^{(2)}).$$

Ideally, one seeks to maximize quality given a budget $B$ on average inference cost:

$$\max_{r \colon \mathcal{X} \to \{0,1\}} \mathbb{E}_{x,y}[Q(x, y, h^{\mathrm{cas}}(x))] \colon \quad \mathbb{E}_x[C(x, h^{\mathrm{cas}}(x))] \leq B. \tag{1}$$

We note that the average cost $\mathbb{E}[C(x, h^{\mathrm{cas}}(x))]$ is related to the *deferral rate* $D(x) = \mathbb{P}(r(x) = 1)$, via $\mathbb{E}[C(x, h^{\mathrm{cas}}(x))] = c^{(1)} + D(x) \cdot c^{(2)}$. In practice, one may set $r(x) = \mathbf{1}(s(x) < t)$ for suitable $s \colon \mathcal{X} \to \mathbb{R}$ and threshold $t \in \mathbb{R}$. One may choose $t$ to satisfy the inference cost constraint.

Now, we discuss cascades for generative LMs. This largely follows the setup described above, except that we now consider *probabilistic* models over *sequences*. Concretely, suppose we have two language models $p^{(1)}, p^{(2)}$, with inference costs $c^{(1)}, c^{(2)}$. Similarly, suppose $q \colon \mathcal{V}^m \times \mathcal{V}^{m'} \times \Delta(\mathcal{V}^n) \to \mathbb{R}_+$ is a measure of the *quality* of a given distribution over responses for a given prompt. A cascade $p^{\mathrm{cas}}$ of these models is parameterized by a deferral rule $r \colon \mathcal{V}^m \to \{0, 1\}$, and is given by:

$$p^{\mathrm{cas}}(\cdot \mid \boldsymbol{x}) = \mathbf{1}(r(\boldsymbol{x}) = 0) \cdot p^{(1)}(\cdot \mid \boldsymbol{x}) + \mathbf{1}(r(\boldsymbol{x}) = 1) \cdot p^{(2)}(\cdot \mid \boldsymbol{x}).$$

Given an input sequence $\boldsymbol{x}$ and target sequence $\boldsymbol{y}$, an LM cascade results in quality and cost

$$Q(\boldsymbol{x}, \boldsymbol{y}, p^{\mathrm{cas}}(\cdot \mid \boldsymbol{x})) \doteq \mathbf{1}(r(\boldsymbol{x}) = 0) \cdot q(\boldsymbol{x}, \boldsymbol{y}, p^{(1)}(\cdot \mid \boldsymbol{x})) + \mathbf{1}(r(\boldsymbol{x}) = 1) \cdot q(\boldsymbol{x}, \boldsymbol{y}, p^{(2)}(\cdot \mid \boldsymbol{x}))$$
$$C(\boldsymbol{x}, p^{\mathrm{cas}}(\cdot \mid \boldsymbol{x})) \doteq \mathbf{1}(r(\boldsymbol{x}) = 0) \cdot c^{(1)} + \mathbf{1}(r(\boldsymbol{x}) = 1) \cdot (c^{(1)} + c^{(2)}).$$

With these, we may construct a similar constrained objective as in Equation 1. Similarly to the classification case, we may parameterize the deferral rule as $r(\boldsymbol{x}) = \mathbf{1}(s(\boldsymbol{x}) < t))$ for a suitable *deferral score function* $s \colon \mathcal{V}^m \to \mathbb{R}$ (which may depend on the output of $p^{(1)}(\cdot \mid \boldsymbol{x})$). We will investigate and analyze different types of deferral score functions on different NLP tasks.

Recently, Chen et al. (2023b) introduced the FrugalGPT system to achieve efficient inference via multiple strategies, including LM cascades. They also learn a deferral score to determine whether or not to terminate prediction; however, this depends on the input prompt and the generated output *text*, and does not consider the model's token-level uncertainty as we shall explore subsequently. A few works have proposed to learn a router which can decide which model to use amongst a set of models depending upon the input prompt (Shnitzer et al., 2023; Hari & Thomson, 2023). However, their settings do not necessarily consider models of increasing capacity and hence, their routers depend only on the input prompt not on the model confidence.

## 3 CONFIDENCE MEASURES FOR LANGUAGE MODEL CASCADES

A key question in the design of cascades is the choice of deferral rule. In this work, we seek to understand the behaviors of different types of deferral functions on NLP tasks. We start by discussing a few natural extensions of commonly used deferral rules for classification.

### 3.1 CHOW-SUM AND CHOW-AVERAGE

**Chow-Sum**. We start with the multi-class classification setting where the output space $\mathcal{Y} = \{1, \ldots, L\}$ and $L \in \mathbb{N}_+$. In the simplest case, one may defer if the *confidence* in the prediction $h^{(1)}(x)$ of the small model is sufficiently low. There are several means of quantifying confidence in classification (Shafer & Vovk, 2008; Guo et al., 2017; Kendall & Gal, 2017; Jiang et al., 2018), but arguably the simplest is the *predicted class probability* (Huang et al., 2018; Wang et al., 2022; Jitkrittum et al., 2023), which aligns with *Chow's rule* from the closely related problem (see Mozannar & Sontag (2020); Narasimhan et al. (2022)) of learning to reject (Chow, 1970):

$$s(x) \doteq p^{(1)}(\hat{y} \mid x), \tag{2}$$

where $p^{(1)}(\cdot \mid x)$ denotes the predictive distribution over possible labels of the small model, and $\hat{y} = \arg\max_{y \in \mathcal{Y}} p^{(1)}(y \mid x)$ denotes the predicted label.

To design a deferral rule for LM cascading, a natural starting point is to mimic the predicted class probability (Equation 2): we may compute the (log) probability of the model generated sequence $\hat{\boldsymbol{y}}$,

$$s_{\mathrm{sum}}(\boldsymbol{x}) \doteq \log p^{(1)}(\hat{\boldsymbol{y}} \mid \boldsymbol{x}) \tag{3}$$

$$= \sum_{i=0}^{|\hat{\boldsymbol{y}}|-1} \log p^{(1)}(\hat{y}'_{i+1} \mid \boldsymbol{x}, \hat{y}'_1, \ldots, \hat{y}'_i), \tag{4}$$

We term this approach Chow-Sum, as it involves the sum of per-token log probabilities. Analogous to the prediction rule for classification, we may set $\hat{\boldsymbol{y}} \doteq \arg\max_{\boldsymbol{y} \in \mathcal{V}^*} p^{(1)}(\boldsymbol{y} \mid \boldsymbol{x})$, denoting by $\mathcal{V}^*$ the set of all sequences. This requires searching over an exponentially large set; however, efficient approximations via greedy or beam search are feasible.

**Chow-Average**. Chow-Sum computes the aggregate sequence-level log-probability. A natural variant is the average of the per-token log-probabilities. This is equivalently the *length normalized* log-probability, or the log-perplexity (Chen et al., 1998):

$$s_{\mathrm{avg}}(\boldsymbol{x}) \doteq \frac{1}{|\hat{\boldsymbol{y}}|} \sum_{i=0}^{|\hat{\boldsymbol{y}}|-1} \log p^{(1)}(\hat{y}'_{i+1} \mid \boldsymbol{x}, \hat{y}'_1, \ldots, \hat{y}'_i). \tag{5}$$

Note that $\hat{\boldsymbol{y}}$ may be computed as above, without incorporating any length-normalization.

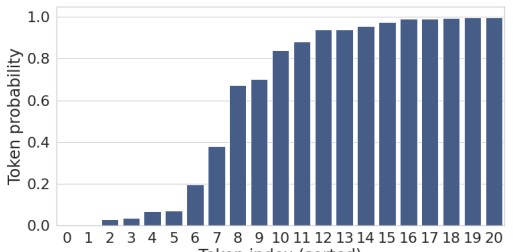

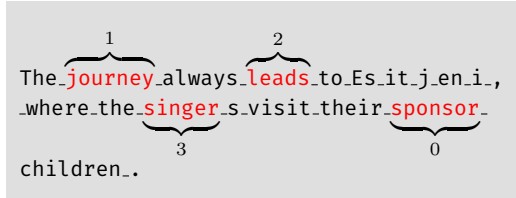

**Model output tokens:**

Figure 2: Example of tokenized FLAN-T5 Base model output on WMT FR → EN. Red tokens have a significantly higher uncertainty compared to the others, as shown in the left plot. (For each red token, we note the rank of its uncertainty score in the right plot.) However, due to the large number of other more predictable tokens, Chow-Sum gives the output a relatively high score.

### 3.2 LIMITATIONS OF CHOW-SUM AND CHOW-AVERAGE

Given that Chow-Sum tracks closely with the well-established Equation 2, it is tempting to conclude that this emphatically solves the LM cascade problem. However, LMs can be susceptible to the *length bias* problem (Murray & Chiang, 2018; Adiwardana et al., 2020): shorter, lower quality responses may receive a higher probability than longer, higher quality responses. This may be seen as a consequence of the fact that each $p(\cdot \mid \boldsymbol{x}, y_1, \ldots, y_i)$ provides an *imperfect* estimate of a "true" probability $p^*(\cdot \mid \boldsymbol{x}, y_1, \ldots, y_i)$, and that errors in these estimates compound with sequence length.

The length-bias issue naturally suggests using an average instead of sum of log-probabilities. However, Chow-Average can over-correct for this length bias, and preferentially defer *shorter* predictions. We will see concrete examples of this behavior in §3.5.

More fundamentally, both approaches are inherently limited in the way they aggregate token-level uncertainty. In particular, computing the sum or average of per-token probabilities may mask settings where *individual tokens* are highly uncertain, even if the entire sequence has reasonably high probability. Such token-level uncertainty may be highly important in certain settings such as fact-answering: here, an LM may be (correctly) highly confident on articles and other grammatical primitives, but these are of less interest than confidence on tokens corresponding to entities (say). This observation has been previously noted and exploited to allow certain "easy" tokens to be quickly decoded (Schuster et al., 2022). This observation has also been exploited in knowledge distillation by using different teaching modes for "easy" versus "hard" tokens (Zhong et al., 2024).

Figure 2 presents an example of this phenomenon on the WMT FR → EN dataset (details in §3.5): there can be cases where most tokens are highly predictable (i.e., $p(y_i' \mid x, y_1', \ldots, y_{i-1}') \approx 1$), but a few tokens are less predictable (i.e., $p(y_i' \mid x, y_1', \ldots, y_{i-1}') \approx 0$). In such cases, Chow-Sum can yield overly optimistic uncertainty estimates. This motivates us to consider richer representations of uncertainty which can capture token-level uncertainty, instead of simply computing the sum or average over the sequence.

### 3.3 BEYOND CHOW-SUM AND CHOW-AVERAGE: CHOW-QUANTILE

The discussion in §3.2 suggests there is value in considering the following generalization of the maximal sequence probability:

$$s_{\text{quant}}(\boldsymbol{x}, \alpha) \doteq \texttt{quantile}_\alpha(p^{(1)}(\hat{y}_1' \mid \boldsymbol{x}), p(\hat{y}_2' \mid \boldsymbol{x}, \hat{y}_1'), \ldots, p(\hat{y}_n' \mid \boldsymbol{x}, \hat{y}_1', \ldots, \hat{y}_{n-1}')),$$

where $\hat{\boldsymbol{y}} \doteq \arg\max_{\boldsymbol{y} \in \mathcal{V}^*} p^{(1)}(\boldsymbol{y} \mid \boldsymbol{x})$ is the most probable output sequence (or a suitable approximation thereof) under $p^{(1)}$. Here, $\texttt{quantile}_\alpha$ computes the $\alpha$-quantile of the set of per-token log probabilities. For instance, $\alpha = 0$ would correspond to taking the minimum per-token log probability as the deferral score. One may regard quantiles as another way of converting the token-level uncertainty distribution into a single score, which are capable of capturing richer information from the token-level uncertainty distribution. For example, employing the *maximal* token uncertainty (i.e., the *minimum* of the per-token probabilities (Stengel-Eskin & Van Durme, 2022)) can be useful in scenarios where most tokens are predictable, but a few important tokens are not (per Figure 2).

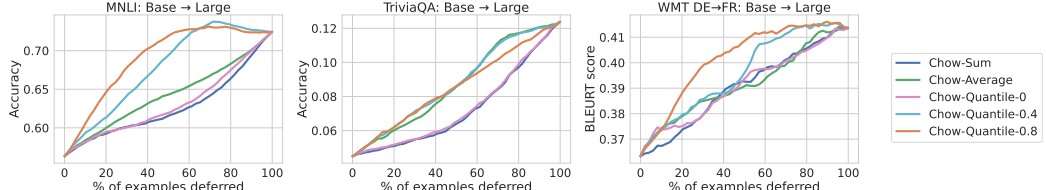

Figure 3: Deferral curves on MNLI, TriviaQA, and WMT DE → FR for a FLAN-T5 Base → Large cascade. `Chow-Quantile` consistently outperforms `Chow-Sum` and `Chow-Average`. This confirms there is value in going beyond naïve sequence probability as an uncertainty measure for cascading.

Next, we evaluate all the aforementioned approaches on multiple NLP tasks. For that, we describe the experimental setup used for the evaluation.

## 3.4 EXPERIMENTAL SETUP

**Models**. We employ FLAN-T5 (Chung et al., 2022) models, which are T5 models (Raffel et al., 2020) that have undergone instruction tuning (Wei et al., 2022). This family offers a range of models of different sizes, spanning Small (80M parameters) to XXL (11B parameters), and have demonstrated strong few-shot performance on a range of NLP benchmarks. In the body, we primarily focus on a two-model cascade of FLAN-T5 Base and FLAN-T5 Large. Results for other models are included in the Appendix. We employ these models with few-shot prompting and greedy decoding.

**Evaluation**. We summarize performance using the *deferral curve*. Consider a candidate deferral rule produced by thresholding $s(\boldsymbol{x}) \in \mathbb{R}$ via $r(\boldsymbol{x}) = \mathbf{1}(s(\boldsymbol{x}) < t)$. Let $p^{\text{cas}}$ denote the associated cascaded LM. For a fixed threshold $t$, we may compute the associated *deferral rate* $\mathbb{P}(r(\boldsymbol{x}) = 1)$, and the associated *cascade quality* $\mathbb{E}[Q(\boldsymbol{x}, \boldsymbol{y}, p^{\text{cas}}(\cdot \mid \boldsymbol{x}))]$. The deferral curve is produced by plotting the trade-off between deferral rate and cascade quality as $t$ is varied. As a scalar summary, we report the *area under the deferral curve* (AUC-DF). For a given dataset, higher AUC-DF values indicate better deferral curves. Note that the range of AUC-DF values vary across datasets, however.

**Datasets**. In the body, we show deferral curves for three different NLP tasks: MNLI (Williams et al., 2018), a multi-class classification problem; TriviaQA (Joshi et al., 2017), a closed-book question answering problem; and WMT DE → FR, a translation problem. We report AUC-DF numbers for an expanded dataset pool. These span *Classification* (IMDb (Maas et al., 2011), SuperGLUE (Wang et al., 2019a), MNLI (Williams et al., 2018), ANLI (Nie et al., 2020)); *Question answering* (TriviaQA (Joshi et al., 2017), NaturalQA (Kwiatkowski et al., 2019), TyDiQA { ID, SW, FI } (Clark et al., 2020)); *Reading comprehension* (Lambada (Paperno et al., 2016), SQuAD (Rajpurkar et al., 2016)); *Translation* (WMT 14: EN → FR (Bojar et al., 2014), WMT 19: DE → FR (Foundation), and WMT 14: FR → EN (Bojar et al., 2014)); and *Common-sense reasoning* (Winogrande (Sakaguchi et al., 2021)). Note that we treat all problems as finding a text to text mapping. So for classification tasks, we encode the classes as strings. For evaluation, we take the model's output text and perform a string comparison to the label. See Table 2 (Appendix) for more details.

## 3.5 EVALUATING CONFIDENCE MEASURES FOR CASCADES

We now empirically validate the critiques in §3.2, demonstrating that using the standard sequence probability (`Chow-Sum`) to defer can result in overly penalizing longer sequences. Moreover, `Chow-Average` flips this bias and overly defers shorter sequences. We then verify that the `Chow-Quantile` generalization proposed above can capture richer token-level uncertainty.

**Summary of results**. Figure 3 plots the deferral curves for three datasets. We see that (for a particular choice of quantile), `Chow-Quantile` consistently outperforms standard `Chow-Sum` and `Chow-Average`. AUC-DF values for all datasets are included in Table 4. Looking at the table, we see that while a *particular* choice of quantile is able to do well, there is no single consistent choice that performs well across tasks. Next, we discuss insights into the results by using WMT FR→EN as an example dataset.

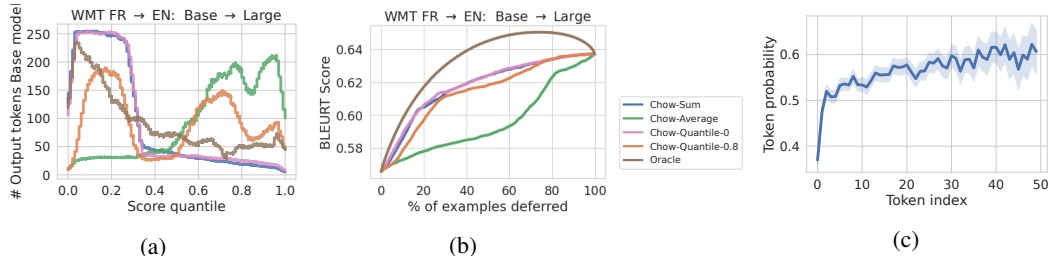

(a)                                    (b)                                    (c)

Figure 4: **(a)** Relation between deferral rules and output length (number of tokens) for WMT FR → EN dataset and FLAN-T5 Base Model. `Chow-Sum` tends to defer longer prompts: the prompts with lowest scores have notably higher length than those with higher scores. Interestingly, `Chow-Average` *over*-corrects this bias: it tends to overly defer prompts with *lower* length. `Chow-Quantile-0` again defers longer outputs more whereas `Chow-Quantile-0.8` initially focuses more on the shorter outputs. Oracle refers to deferring using the difference of BLEURT scores of the two models. Oracle also tends to defer longer outputs, but the preference is moderate as compared to `Chow-Sum`. **(b)** Corresponding deferral curves. **(c)** Analysis of token-level uncertainty on WMT FR → EN. For each token index $i$, the corresponding average prediction probability across all examples (with prediction length $\leq i$) for FLAN-T5 Base. We observe that later tokens tend to have higher probability, i.e., the model is generally the most uncertain for early tokens.

| | |
|---|---|
| For the Australian Government, Keith Brown called Mr. Carmichael "unjustly" to support the inclusion of the Ecosse in the HS2.net network. ?? ?? ?? ?? ?? ?? ?? ?? ?? .... | The announcement of the release of a new album by David Bowie has left everyone a little bit a little a little a little a little a little bit a little bit a little bit a little bit .... |
| The lyric Adéro, a concert-recording held last August, was added to the repertoire, with the competition of the coloratura soprano Marie- ?? ve Munger. The lyric Adéro was added to the repertoire in August. The lyric Adéro was added to the repertoire in August. .... | "The Emergency Room at the Hotel-Dieu must be closed as soon as possible, and for us, it is November 4th," says Lo ?? c Capron, Chairperson of the APHP Medical Committee (MCM), who supports the direction. |
| Obama lays out his reform plan | A memorial ceremony for the victims |

Figure 5: FLAN-T5 Base predictions on WMT FR → EN. **Top:** Predictions with the longest lengths. These tend to have repetitions, indicating low quality output that could be resolved with a larger model; length does have some signal in identification of good candidates for deferral. **Middle:** The predictions which `Chow-Quantile-0` tends to defer. This quantile tends to identify repetitions and "??" (unknown tokens) as these tokens tend to have lower probability. **Bottom:** The predictions which `Chow-Quantile-0.8` tends to defer. This quantile prioritizes deferring shorter inputs.

**Why can Chow-Sum and Chow-Average be sub-optimal?** To better understand the reason for the sub-optimality of `Chow-Sum`, Figure 4 studies the relation between the deferral rule and output length. Specifically, for each test prompt $x$, let $\hat{y}$ denote the result of decoding via the small model in the cascade. For each deferral rule, we compute the corresponding score $s(x)$ and the length $|\hat{y}|$. For ease of comparison across different rules, we convert each score into the corresponding quantile.

Figure 4 reveals that `Chow-Sum` tends to defer prompts with larger output lengths: the prompts with lowest scores have notably higher output length than those with higher scores. This makes us ask if it is all bad to defer prompts with longer outputs? We observe that for the Base model on the WMT datasets, even the BLEURT (Sellam et al., 2020) scores tend to have a non-zero negative correlation with output lengths (Table 3, Appendix). A closer look at the model predictions shows that longer predictions tend to have repetitions, as shown in Figure 5 (Top) and hence, are good candidates for deferral. (The shown predictions are truncated for clarity.)

This shows that there is some signal in output length as a deferral score. However, `Chow-Sum` is overly biased towards deferring longer predictions and hence, can be sub-optimal. Interestingly, `Chow-Average` *over*-corrects this bias: it tends to overly defer prompts with *lower* output length.

**Why does Chow-Quantile help?** As discussed above, `Chow-Quantile` is able to capture rich information from the token-level uncertainty vector. We discuss below why `Chow-Quantile-0` and `Chow-Quantile-0.8` work well with respect to the WMT FR→EN dataset.

`Chow-Quantile-0`: The main insight is that the minimum token probability is able to capture repetitions and "??" (unknown tokens), as they generally tend to have lower probability values and are more uncertain. This confirms our understanding that quantiles can capture richer token-level uncertainty. We show two examples with the minimum `Chow-Quantile-0` value for the WMT FR→EN dataset and FLAN-T5 Base in Figure 5 (Middle).

`Chow-Quantile-0.8`: Interestingly, `Chow-Quantile-0.8` tends to defer shorter predictions. We show two examples with the minimum `Chow-Quantile-0.8` value in Figure 5 (Bottom).

To understand this, Figure 4c shows the average token probability as a function of the token index, for the WMT EN → FR dataset and FLAN-T5 Base model. As the token index increases, the average probability increases; i.e., the model tends to become more confident. Hence, the `Chow-Quantile-0.8` is able to focus more on the shorter, uncertain outputs.

In summary, we have seen that `Chow-Quantile-0` is able to focus more on identifying the presence of repetitions and unknown tokens "??" whereas `Chow-Quantile-0.8` is able to capture the uncertainty in shorter predictions better. Thus, we conclude that different quantiles are able to capture richer and complementary measures of uncertainty. Moreover, we have already seen that there is no one quantile which works well across all datasets. Given this, a natural option is to learn how to combine various quantiles for a given dataset, which we consider next.

## 4 POST-HOC DEFERRAL RULES

We show that training *post-hoc* deferral rules based on probability quantiles, and (optionally) suitable embeddings from the small and large model, can significantly improve the cost-quality tradeoff.

### 4.1 POST-HOC DEFERRAL RULE TRAINING

The idea of learning when to defer in a cascade follows a recent line of work on classification (Narasimhan et al., 2022; Kag et al., 2023; Jitkrittum et al., 2023). In a nutshell, for suitable feature mapping $\Phi(\boldsymbol{x}) \in \mathbb{R}^d$, we seek to learn a deferral score $s(\boldsymbol{x}) \in \mathbb{R}$ via a standard model class (e.g., a feedforward network). We then defer using $r(\boldsymbol{x}) = \mathbf{1}(s(\boldsymbol{x}) < t)$.

To construct the input features, we set $\Phi(\boldsymbol{x})$ to be a fixed length vector comprising the per-token probability quantiles from the small model. Additionally, we add the aggregate scores from `Chow-Sum` and `Chow-Average` (see Appendix F). To fit the deferral scorer on a training set of input prompts $\{\boldsymbol{x}_i\}$, we minimize an empirical loss against a set of target labels $\{z_i\}$. For tasks based on accuracy, we set $z_i = 1$ iff the large model is correct, and the small model is incorrect on the given example; i.e., it would benefit to defer to the larger model. We then fit the scorer with the binary logistic loss. For translation tasks, the target is the difference of BLEURT scores of the two models; we train with the square loss. We call this method `Post-Hoc-Quantile` (see Appendix D for details).

### 4.2 LEVERAGING INTERMEDIATE EMBEDDINGS

The above target labels exploit information from the large model during *training*. Importantly, we cannot directly use such information during *inference*, as it would require querying the large model (and thus defeat the point of cascading). Note, however, that in some settings it may be feasible to use *intermediate* information from the large model, e.g., token embeddings from an intermediate layer. Prior work has noted that such intermediate embeddings can often contain valuable information by themselves (Schuster et al., 2022).

Inspired by this, we thus study the viability of using such intermediate embeddings for training post-hoc deferral rules. For encoder-decoder models such as T5, such embeddings can be from either the encoder, decoder, or both. We study two methods - one which uses the final decoder embeddings of the smaller model averaged over all tokens. We call this method `Post-Hoc-Embed-1`. In the second method, we add the first token embedding from the first decoder layer of the large model as another input to the post-hoc rule. We call this method `Post-Hoc-Embed-1+2`.

| Dataset | Chow-Sum | Chow-Average | Chow-Quantile-0 | Chow-Quantile-0.4 | Chow-Quantile-0.8 | Post-Hoc-Quantile | Post-Hoc-Embed-1+2 |
|---|---|---|---|---|---|---|---|
| ANLI-R1 | 0.524 (+4.59) | 0.519 (+3.59) | **0.534 (+6.58)** | 0.515 (+2.79) | 0.515 (+2.79) | **0.534 (+6.58)** | **0.563 (+12.51)** |
| ANLI-R2 | 0.446 (+0.67) | 0.446 (+0.67) | **0.449 (+1.35)** | 0.440 (-0.67) | 0.441 (-0.45) | 0.442 (-0.22) | **0.462 (+4.36)** |
| ANLI-R3 | 0.413 (+3.50) | 0.422 (+5.76) | 0.417 (+4.51) | **0.425 (+6.51)** | 0.424 (+6.26) | **0.425 (+6.51)** | **0.440 (+10.45)** |
| BoolQ | 0.838 (+2.57) | 0.838 (+2.57) | 0.838 (+2.57) | 0.838 (+2.57) | 0.838 (+2.57) | **0.840 (+2.81)** | **0.854 (+4.64)** |
| IMDb | **0.964 (+1.15)** | **0.964 (+1.15)** | **0.964 (+1.15)** | **0.964 (+1.15)** | **0.964 (+1.15)** | **0.964 (+1.15)** | **0.965 (+1.31)** |
| Lambada | **0.703 (+3.38)** | 0.702 (+3.23) | **0.703 (+3.38)** | 0.694 (+2.05) | 0.687 (+1.02) | 0.701 (+3.08) | 0.692 (+1.82) |
| MNLI | 0.627 (-2.63) | 0.642 (-0.31) | 0.631 (-2.01) | 0.677 (+5.12) | 0.691 (+7.29) | **0.711 (+10.4)** | **0.722 (+12.24)** |
| PIQA | 0.712 (+0.99) | 0.708 (+0.42) | 0.715 (1.41) | 0.706 (+0.14) | 0.705 (+0.00) | **0.717 (+1.70)** | **0.727 (+3.26)** |
| SQuaD | 0.410 (+2.24) | 0.408 (+1.74) | **0.410 (+2.24)** | 0.399 (-0.49) | 0.398 (-0.74) | 0.409 (+1.99) | **0.412 (+2.76)** |
| TriviaQA | 0.073 (-13.09) | 0.087 (+3.57) | 0.073 (-13.09) | 0.088 (+4.76) | 0.086 (+2.38) | **0.097 (+15.47)** | **0.100 (+19.56)** |
| TyDiQA-FI | **0.332 (+3.10)** | 0.322 (+0.00) | 0.328 (+1.86) | 0.318 (-1.24) | 0.312 (-3.10) | 0.331 (+2.79) | **0.338 (+5.27)** |
| TyDiQA-ID | **0.247 (+7.39)** | 0.239 (+3.91) | 0.245 (+6.52) | 0.230 (+0.00) | 0.230 (+0.00) | 0.235 (+2.17) | 0.242 (+5.35) |
| TyDiQA-SW | **0.170 (+9.67)** | 0.147 (-5.16) | 0.166 (+7.09) | 0.141 (-9.03) | 0.140 (-9.67) | 0.163 (+5.16) | 0.162 (+4.98) |
| Winogrande | 0.648 (+2.36) | 0.648 (+2.36) | 0.648 (+2.36) | **0.649 (+2.52)** | 0.648 (+2.36) | 0.648 (+2.36) | **0.670 (+5.87)** |
| WMT DE → FR | 0.390 (+0.00) | 0.392 (+0.51) | 0.391 (+0.25) | 0.396 (+1.53) | 0.401 (+2.82) | **0.404 (+3.58)** | **0.404 (+3.69)** |
| WMT EN → FR | 0.436 (+1.16) | 0.435 (+0.92) | 0.436 (+1.16) | 0.435 (+0.92) | 0.439 (+1.85) | **0.441 (+2.32)** | **0.444 (+3.05)** |
| WMT FR → EN | 0.618 (+2.82) | 0.595 (-0.99) | 0.618 (+2.82) | 0.595 (-0.99) | 0.614 (+2.16) | **0.619 (+2.99)** | 0.618 (+2.89) |

Table 1: Table showing area under the deferral curve (AUC-DF). Numbers in brackets indicate % change over the random baseline. The post-hoc deferral rule approach is the most consistently performant method. We use bold & black to denote the best method amongst Chow-* and Post-Hoc-Quantile methods. We color Post-Hoc-Embed-1+2 with blue if it is the best amongst all.

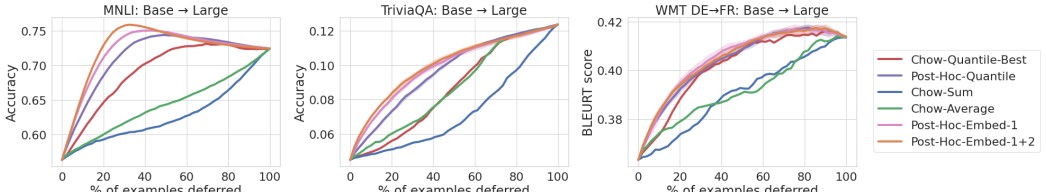

Figure 6: Deferral curves on MNLI, TriviaQA, and WMT DE → FR for a FLAN-T5 Base → Large cascade. The post-hoc deferral rule yields consistently on-par or superior performance compared to other methods. Further exploiting intermediate embeddings of the large model yields large gains. See Table 4 for a summary of deferral curves on multiple datasets.

We remark that previous work (Ren et al., 2023) has shown the value of final layer embeddings for *selective generation*, and the related problem of *out-of-distribution* detection. We caution also that while not as expensive as querying the entire large model, even extracting intermediate decoder embeddings can involve a non-trivial cost. Nonetheless, in settings where some increase in cost is acceptable, it is of interest whether these embeddings offer significantly valuable information.

## 4.3 HOW WELL DOES POST-HOC DEFERRAL WORK?

For the same experimental setup as the previous section, Table 4 summarizes the area under the deferral curve (AUC-DF) across various datasets. Numbers are averaged over 5 random runs. We see that the post-hoc deferral rule approach consistently performs the best; in scenarios where other methods are better, the difference is minimal. For a more fine-grained analysis, Figure 6 plots the full deferral curves on MNLI, TriviaQA, and WMT. In the plot, Chow-Quantile-Best chooses the best method out of Chow-Quantile-* based on the validation data. We see that post-hoc deferral rules are generally on-par with the Chow-* family of (non-learned) deferral rules.

We see that Post-Hoc-Embed-1 is able to improve upon Post-Hoc-Quantile and Chow-Quantile-* methods slightly but is slightly inferior compared to the Post-Hoc-Embed-1+2 method. This intuitively makes sense as this has more information compared to the Post-Hoc-Quantile method but still does not include any information about model 2.

Strikingly, further using the larger model's intermediate embeddings can lead to significant improvements across all deferral rates, particularly for *classification tasks*. Intuitively, the first token's intermediate embedding could have a lot of information for classification tasks, where the answers typically comprise of a single word and only a couple of tokens. However, for generation and translation tasks with longer outputs, the main token containing the answer could be present later in the sequence and thus, there may be limited use of using the first token embedding. One may wonder if we really need quantiles to train the post-hoc deferral rule. In Appendix E.1, we show that naïvely passing the probability vector with padded zeros performs poorly in many cases.

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

# Language Model Cascades:
# Token-level uncertainty and beyond

## Appendix

## A DISCUSSION AND FUTURE WORK

We have seen that there is value in going beyond simple sequence level uncertainty and considering finer measures of token level uncertainty as deferral rules for cascades. Moreover, we have seen that intermediate embeddings from the larger model can further boost performance.

This work raises a number of interesting directions for future work. First, we have used a 5-layer MLP to train post-hoc deferral rules using the quantiles of the probability distribution. It would be interesting to see how well a 1-layer Transformer works for this task, to potentially better exploit the sequential nature of the token probability vector. We have focused on FLAN-T5 instruction tuned Encoder-Decoder models in this work. We believe that the insights and methods should generalize to Decoder-only architectures. It would be interesting to evaluate the proposed approaches for such architectures. Moreover, it has been observed that models with RLHF finetuning become uncalibrated (Figure 8 in OpenAI (2023)). It would be interesting to see how various finetuning steps affect the findings in this work and what consequences they have for designing efficient cascades.

Multiple works have considered alternative notions of uncertainty using the generative abilities of LMs, for example, reprompting the model to ask how confident it is about the answer or output an additional confidence score as part of the outputs (Kadavath et al., 2022). It would be interesting to evaluate how well these measures work for cascades. See Appendix C for details.

## B LIMITATIONS

One potential limitation of post-hoc deferral rule training is that it may not work well if there is some distribution shift between the probability values in the train and test sets. It also may not work in settings where we have limited training data, especially where we use high dimensional features like embeddings.

## C UNCERTAINTY QUANTIFICATION FOR LMS

There has been a large body of work on uncertainty quantification for LMs. We discuss some of the approaches below. They can be broadly divided into the following categories.

**Consensus-based**. One limitation of Equation 3 is that it considers a *single* output sequence, e.g., the most likely one. However, as there are many sequences that have similar meaning, it is intuitively more reliable to consider drawing *multiple* sequences from $p(\cdot \mid \boldsymbol{x})$. One may then assess the *consensus* in resulting predictions to measure confidence (Wang et al., 2023; Xiao et al., 2021; Chen et al., 2023c); this can help distinguish between models that are locally diffused versus peaked around a candidate sequence. Recently, Yue et al. (2023) explored the use of answer consistency to construct effective cascades.

**Deep ensembles and dropout**. One approach to measure confidence is to create an ensemble of different models, and suitably aggregate their predictions (e.g., based on disagreement) (Van Landeghem et al., 2022; Wang et al., 2019b; Gleave & Irving, 2022). However, these uncertainty estimation procedures involve additional computation (e.g., multiple inferences with a single model in dropout-based approaches and single inference with multiple models in ensemble-based approaches) compared to simply using softmax probability outputs from a single network. Such approaches are less appealing for use in cascades, where the primary goal is to improve efficiency.

**Post-hoc calibration/Answer- and length-bias calibration**. For tasks involving question-answering with multiple choices (e.g., (A), (B), (C)), several works have demonstrated that LMs can have prior biases to certain answers, which can be identified and corrected (Zhao et al., 2021; Holtzman et al., 2021; Kumar, 2022; Murray & Chiang, 2018; Mielke et al., 2022; Jiang et al., 2021).

**Semantic entropy**. Another key challenge in measuring the uncertainty for natural language outputs is that there are a lot of semantic equivalent sentences and hence, the probability can be divided among multiple outputs which mean the exact same thing. Kuhn et al. (2023a) proposes to mitigate this problem by sampling multiple outputs and then clustering semantically equivalent outputs together and combining their probability together. It would be interesting to understand how well this method can work for our setting.

**Generative uncertainty**. The above has largely focussed on generalizing the standard maximum predictive probability (Equation 2) from classification to the LM setting. While this by itself leads to a rich array of possible confidence measures, LMs intriguingly offer a wholly new possible means of assessing confidence: one may directly probe the model to obtain how confident it is on the proposed answer (Kadavath et al., 2022). Kadavath et al. (2022) discuss various ways of the input prompt format for this confidence probe. They also discuss the training of an additional head of the model to predict the model confidence but again, it is not clear how this compares with the standard probability output by the model without any additional finetuning. However, (Shrivastava et al., 2023) found that the confidence measures generated linguistically give worse estimates of uncertainty compared to the classical softmax-based measures even when these softmax-based probabilities come from a different and weaker model. Moreover, they observed that two sources of uncertainty are complementary and it can be beneficial to combine them.

**Other work**. Zhao et al. (2023) proposed sequence-level calibration as a means to improve the generative ability of LMs; such calibration could also be useful in improving methods such as `Chow-Sum`. Kuhn et al. (2023b) proposed to ask the model to detect ambiguous questions which the model is likely to get wrong and answer clarifying questions if the question is indeed ambiguous. Hendy et al. (2023) proposed to use an exogeneous quality estimation model to decide how to route between two models. Šakota et al. (2023) similarly proposed to train a meta-model to pick an appropriate model from a family. Fadeeva et al. (2023) did a comprehensive experimental analysis of various uncertainty methods.

# D EXPERIMENTAL DETAILS

We use a Multi-Layer-Perceptron (MLP) with 5 layers and 32 hidden units per dimension for training post-hoc deferral rules with Quantiles with batch normalization layers. We train for 200 epochs and early stop depending on the AUC of predicting the target label on the validation set for classification tasks and validation regression loss for the regression tasks. Since the embeddings are high dimensional, we use MLP with 2 layers and 8 hidden units to prevent overfitting. We train for the same number of epochs with early stopping. We use ADAM optimizer with a learning rate of $10^{-5}$. We use a batch size of 16 for training. We train over 5 random runs and show mean and standard deviations over the runs.

We limit the number of examples per dataset per split to be 10k. Since many datasets do not have all the three splits available. We split the training set into train (0.8 fraction) and validation sets (0.2 fraction). And, use the validation or test split for reporting numbers. The details of all the datasets are included in Table 2.

| Dataset | Category | # Shots | Train | | Test | | # target tokens | |
|---|---|---|---|---|---|---|---|---|
| | | | Split | Size | Split | Size | Mean | Std |
| BoolQ | Classification | 2 | Train | 9417 | Val | 3270 | 1 | 0 |
| IMDb | Classification | 2 | Train | 9991 | Test | 9981 | 1 | 0 |
| MNLI | Classification | 3 | Train | 10000 | Test | 9832 | 1.9 | 1.4 |
| ANLI-R1 | Classification | 3 | Train | 9999 | Test | 1000 | 1.9 | 1.3 |
| ANLI-R2 | Classification | 3 | Train | 10000 | Test | 1000 | 1.9 | 1.4 |
| ANLI-R3 | Classification | 3 | Train | 10000 | Test | 1200 | 1.9 | 1.3 |
| PIQA | Classification | 2 | Train | 10000 | Val | 1838 | 24.4 | 25.3 |
| Winogrande | Classification | 2 | Train | 9998 | Val | 1267 | 1.3 | 0.7 |
| TriviaQA | Generation | 2 | Train | 10000 | Val | 10000 | 4.0 | 3.3 |
| NaturalQA | Generation | 2 | Train | 10000 | Val | 3610 | 3.5 | 2.1 |
| SQuaD | Generation | 2 | Train | 10000 | Val | 9997 | 4.6 | 4.1 |
| Lambada | Generation | 2 | Train | 4857 | Test | 5152 | 2.0 | 1.1 |
| TyDiQA-FI | Generation | 2 | Train | 3659 | Val | 782 | 14.1 | 27.4 |
| TyDiQA-SW | Generation | 2 | Train | 3611 | Val | 499 | 12.2 | 24.3 |
| TyDiQA-ID | Generation | 2 | Train | 3655 | Val | 565 | 10.3 | 21.0 |
| WMT FR→EN | Translation | 2 | Train | 9976 | Test | 3003 | 35.3 | 40.7 |
| WMT EN→FR | Translation | 2 | Train | 9974 | Test | 3003 | 54.3 | 48.5 |
| WMT DE→FR | Translation | 2 | Train | 9999 | Val | 1497 | 33.6 | 23.2 |

Table 2: The table shows the number of shots used for each dataset, the splits used for training and testing, the number of examples in each split and the average number of target tokens in the train split of each of the datasets.

# E  ADDITIONAL EXPERIMENTAL RESULTS

## E.1  ABLATIONS WITH PROBABILITY VECTORS AS FEATURES

In this section, we show that simply using the sequence of probability values with padded zeros to account for variable length sequences does not work as well as using quantiles. This intuitively makes sense, as the different token uncertainties are not aligned with each other which makes learning hard for the MLP. We further show that using sorted probability values does not work as well as using the quantiles: quantiles help better align the different uncertainty scores for variable length sequences.

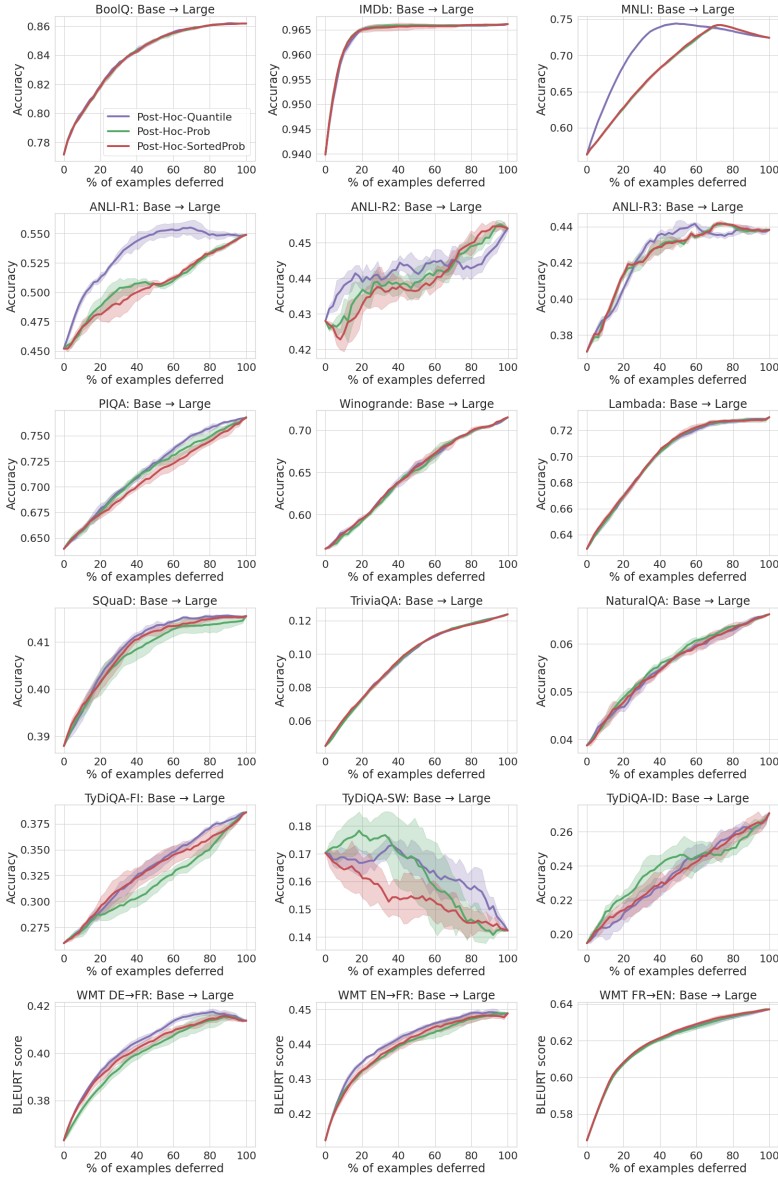

Figure 7: Deferral curves on different datasets for a FLAN-T5 Base → Large cascade. Post-Hoc-Quantile denotes the post-hoc deferral rule which includes quantile features. Post-Hoc-Prob denotes the post-hoc deferral rule which includes the entire probability vector padded with zeros. Post-Hoc-SortedProb denotes the post-hoc deferral rule which includes the sorted probability vector padded with zeros.

| Dataset | Small | Base | Large | XL | XXL |
|---|---|---|---|---|---|
| WMT DE → FR | −0.35 | −0.28 | −0.22 | −0.30 | −0.24 |
| WMT EN → FR | −0.26 | −0.58 | −0.26 | −0.33 | −0.29 |
| WMT FR → EN | −0.16 | −0.20 | −0.21 | −0.24 | −0.10 |

Table 3: Correlation of BLEURT scores with prediction lengths across FLAN-T5 model sizes. We observe that there is a non-zero correlation between output lengths and BLEURT scores. Hence, there is some signal in using output length as a criterion for deferral.

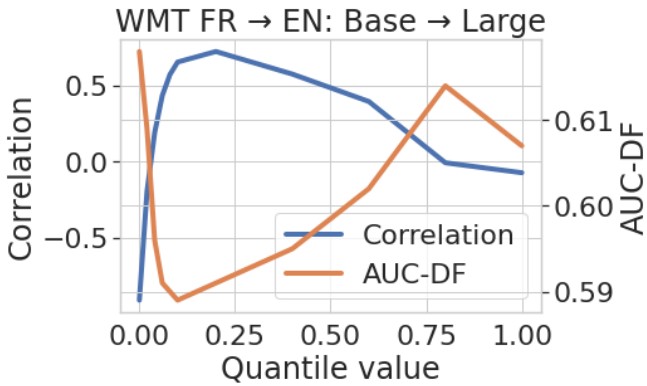

Figure 8: Different quantile scores have different correlations with prediction lengths and different AUC-DF for the deferral curves. For this particular dataset, the best quantile values are 0.0 and 0.8.

## E.2 OUTPUT LENGTH AND BLEURT CORRELATION

Table 3 shows there is a significant (negative) correlation of BLEURT scores with prediction lengths across FLAN T5 model sizes. This verifies there is some signal in using output length as a criterion for deferral.

### E.3 ADDITIONAL RESULTS FOR VARIOUS MODELS AND DATASETS

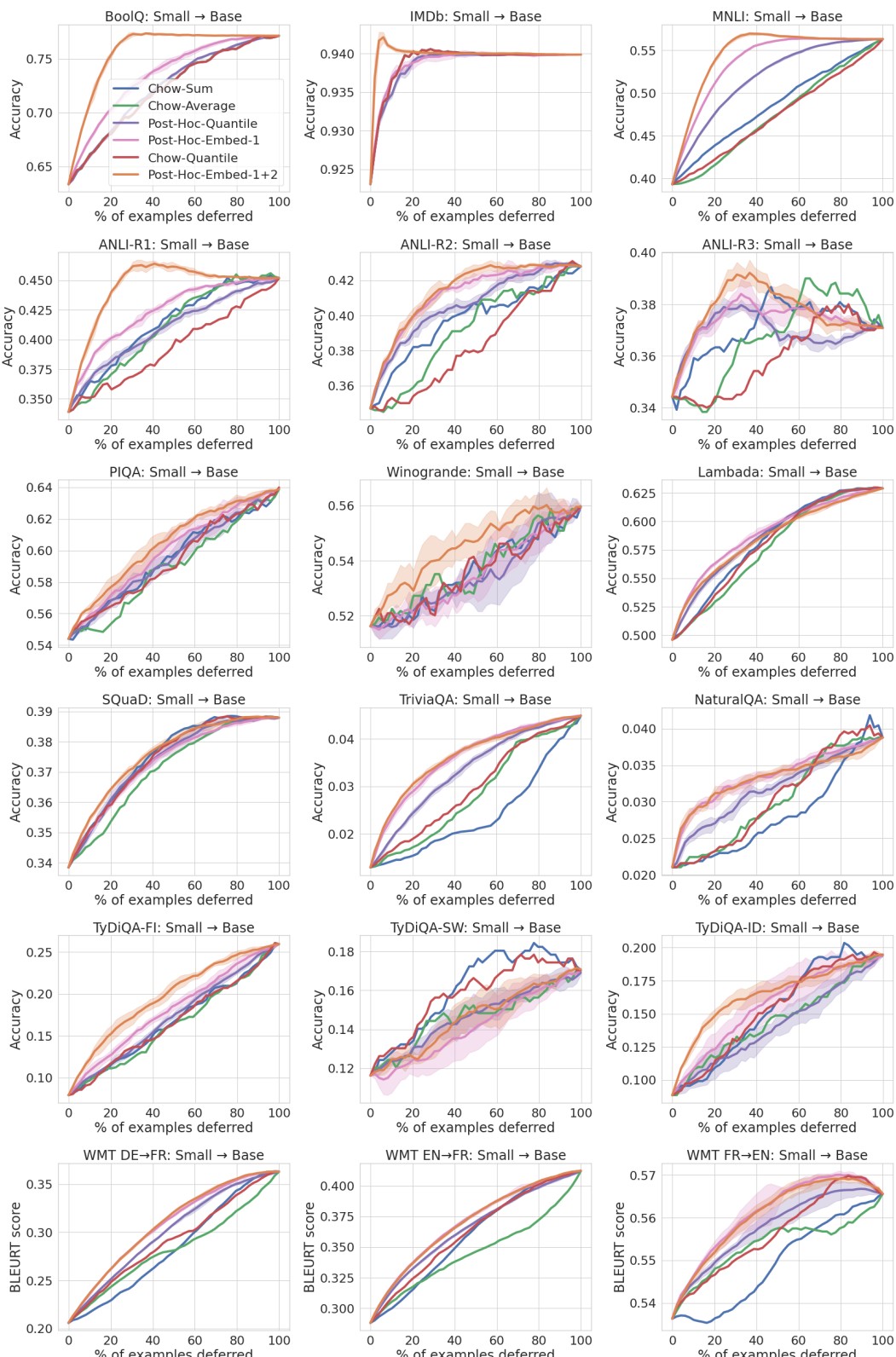

Figure 9: Cascade curves for all datasets and deferral methods for Small → Base model.

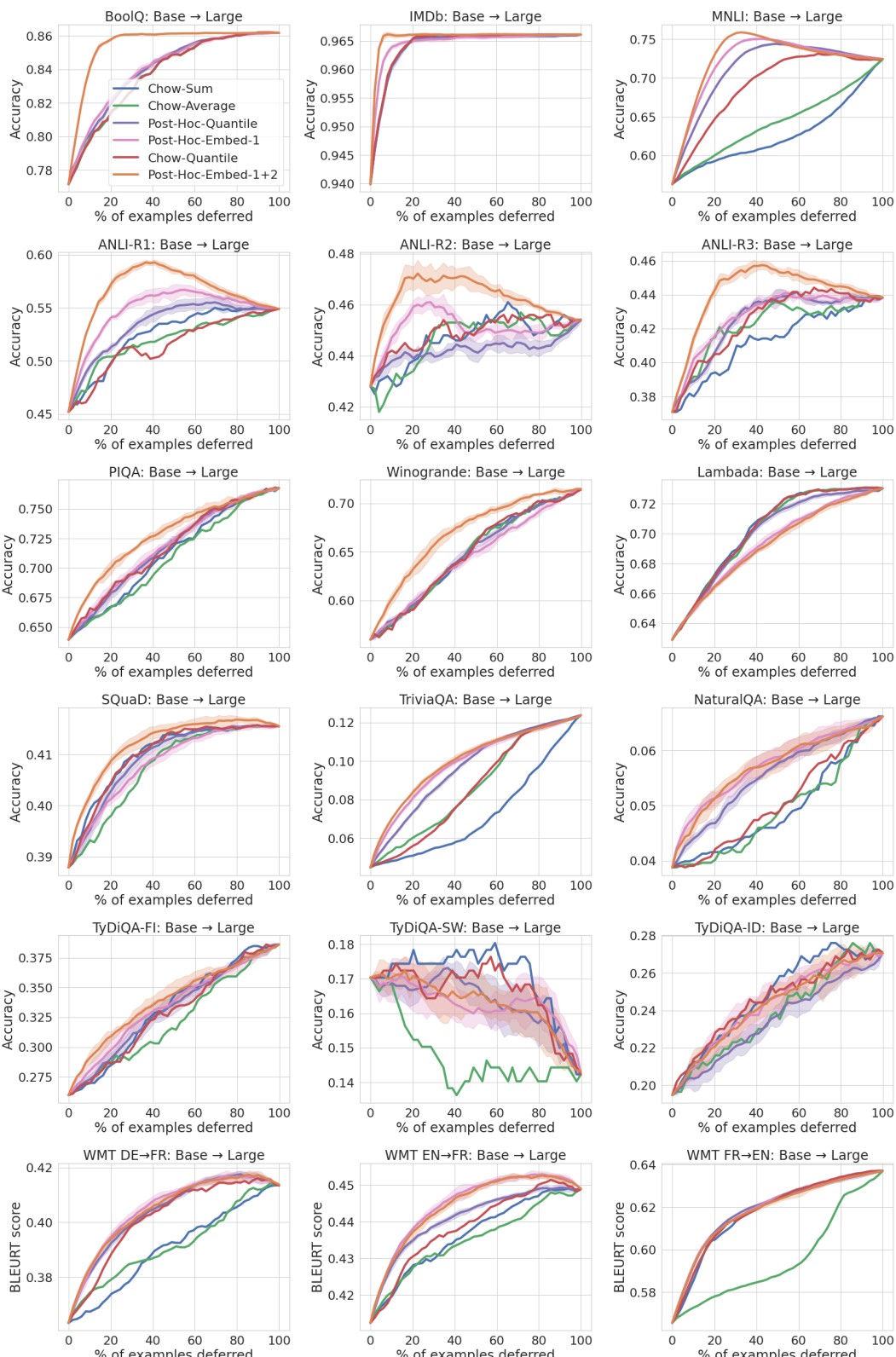

Figure 10: Cascade curves for all datasets and deferral methods for Base → Large model.

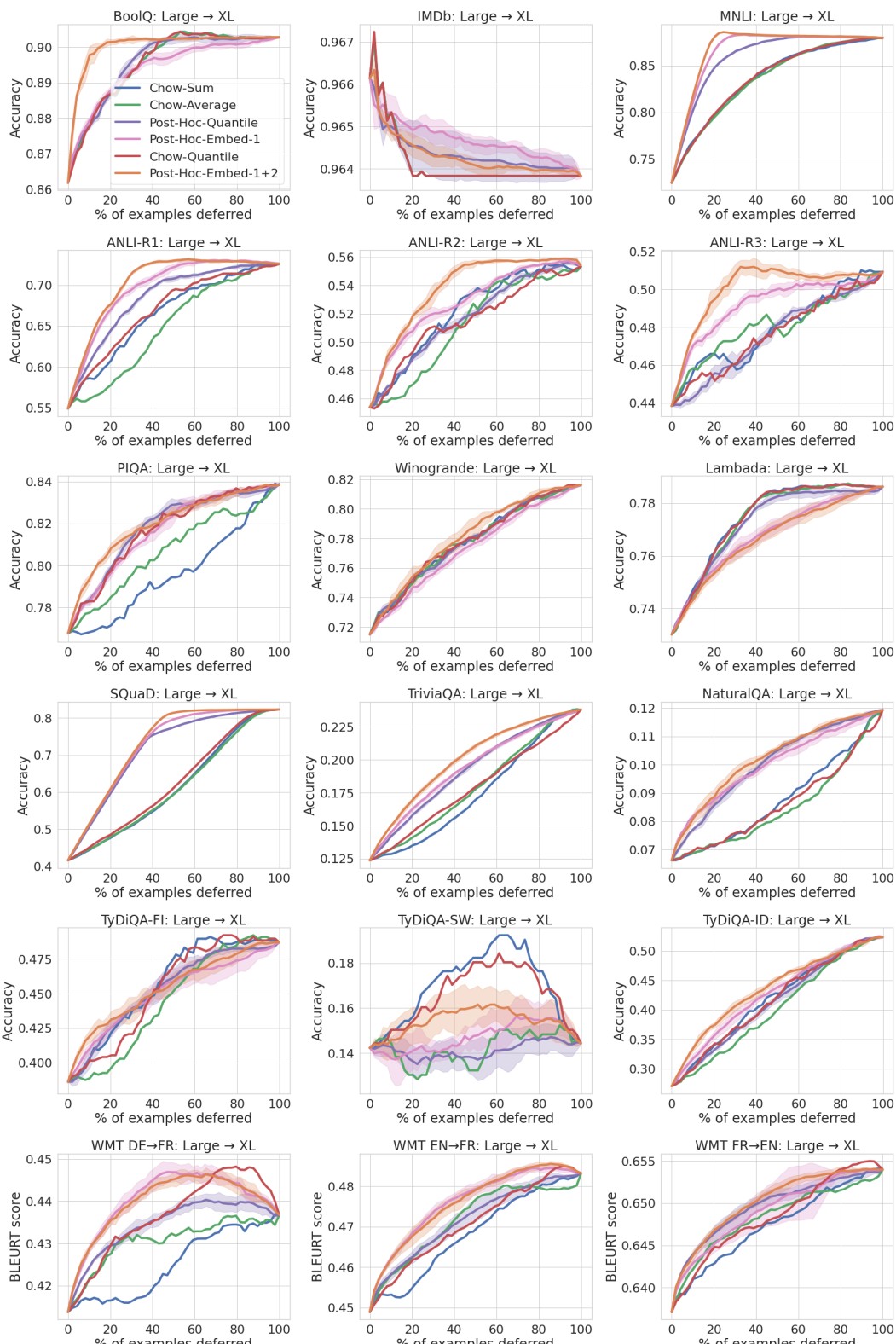

Figure 11: Cascade curves for all datasets and deferral methods for Large → XL model.

### E.4 DIFFERENT VERSION OF LENGTH PLOT

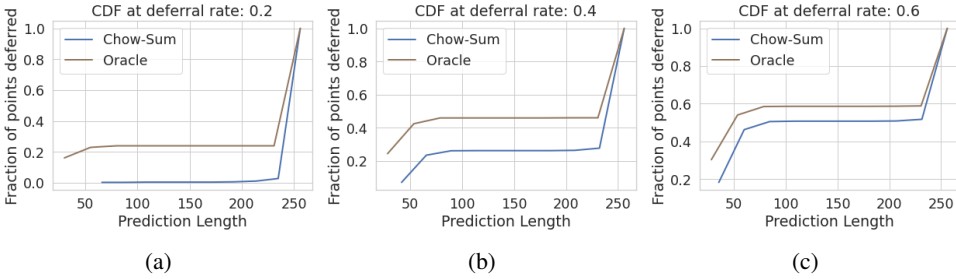

Figure 12: CDF plot for the WMT FR → EN dataset for the FLAN T5 Base → Large model for different deferral rates. Each plot considers the points which are deferred by each method at the mentioned deferral rate. Each point $(x, y)$ represents that $y$ fraction of points at the desired deferral rate have prediction lengths less than $x$. We can see that Oracle has larger fractions of points deferred with smaller prediction length. Hence, Chow-Sum prefers to defer longer predictions as compared to the Oracle method.

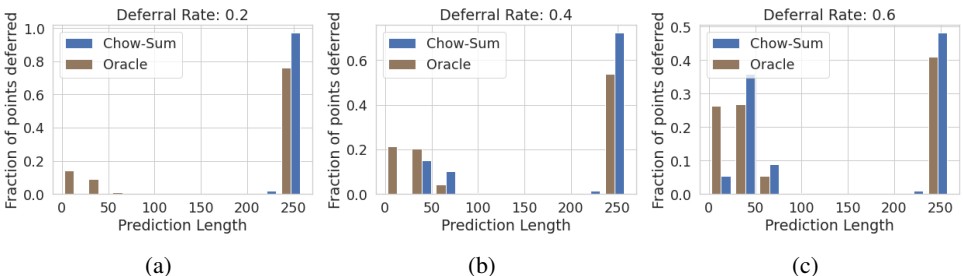

Figure 13: CDF plot for the WMT FR → EN dataset for the FLAN T5 Base → Large model. Each plot considers the points which are deferred by each method at the mentioned deferral rate. We can see that Oracle has larger fractions of points deferred with smaller prediction lengths. Hence, Chow-Sum prefers to defer longer predictions as compared to the Oracle method.

### E.5 CASCADE CURVES WITH RELATIVE LATENCY

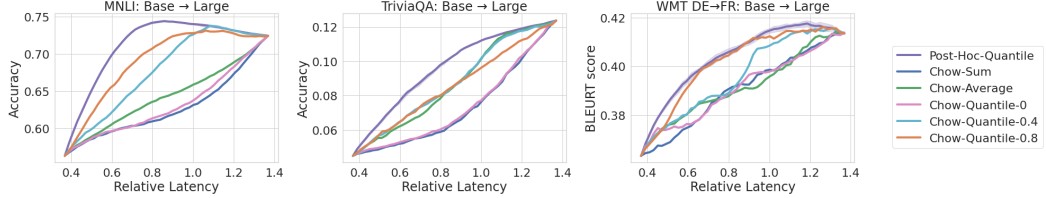

Figure 14: Deferral curves on MNLI, TriviaQA, and WMT DE → FR for a FLAN-T5 Base → Large cascade.

### E.6 RESULTS WITH ENCODER AND DECODER EMBEDDINGS

Post-Hoc-Embed-Dec-1 is the method Post-Hoc-Embed-1 method from main draft which includes decoder embeddings from model 1. Post-Hoc-Embed-Enc+Dec-1 additionally includes encoder embeddings from model 1.

Similarly, Post-Hoc-Embed-Dec-1+2 is the method Post-Hoc-Embed-1+2 method from main draft which includes decoder embeddings from model 1 and intermediate decoder embeddings from model 1. Post-Hoc-Embed-Enc+Dec-1+2 additionally includes encoder embeddings from model 2.

We do not observe any additional gains from including encoder embeddings in these methods.

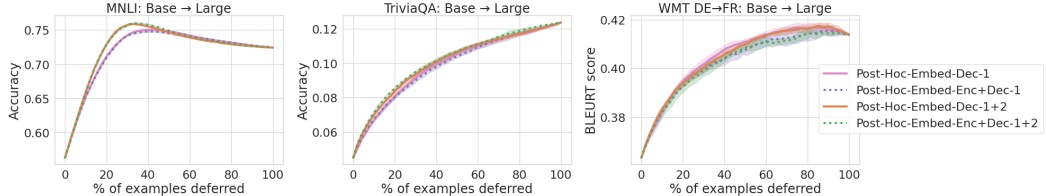

Figure 15: Deferral curves on MNLI, TriviaQA, and WMT DE → FR for a FLAN-T5 Base → Large cascade with additional encoder embeddings.

## E.7 RESULTS WITH BEAM SEARCH DECODING

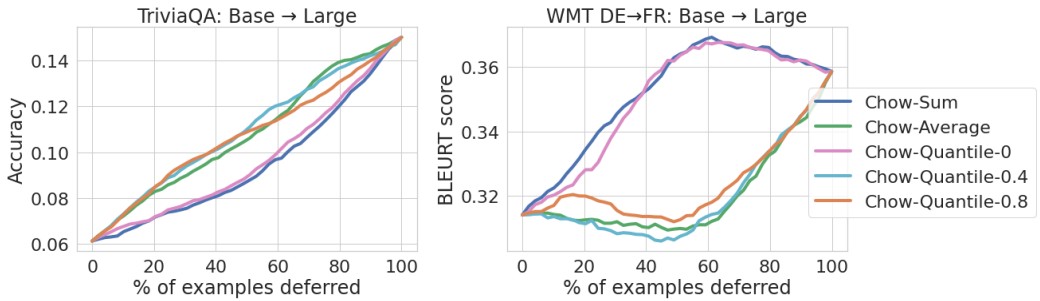

Figure 16: Deferral curves on TriviaQA, and WMT DE → FR for a FLAN-T5 Base → Large cascade with beam search decoding.

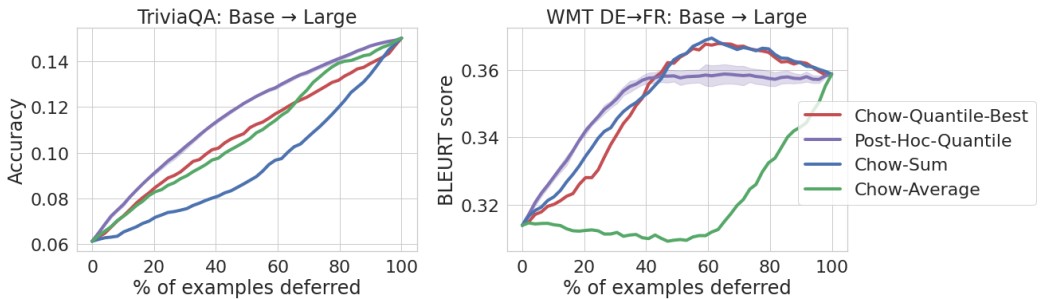

Figure 17: Deferral curves on TriviaQA, and WMT DE → FR for a FLAN-T5 Base → Large cascade with beam search decoding.

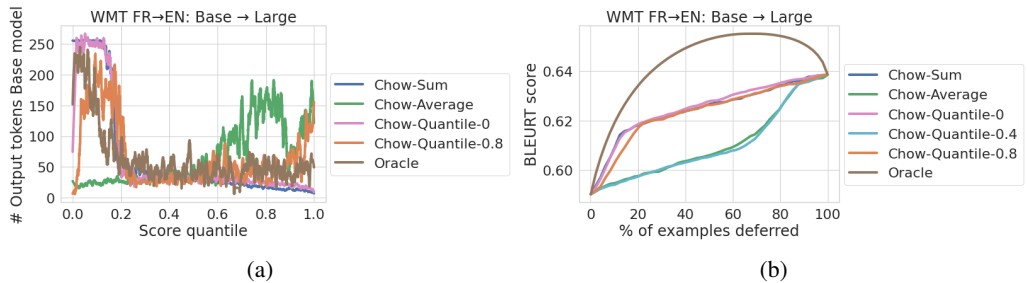

(a)                  (b)

Figure 18: **Left**: Relation between deferral rules and output length (number of tokens) for WMT FR → EN dataset and FLAN-T5 Base Model with decoding using beam search. `Chow-Sum` tends to defer longer prompts: the prompts with lowest scores have notably higher length than those with higher scores. Interestingly, `Chow-Average` *over*-corrects this bias: it tends to overly defer prompts with *lower* length. Oracle refers to deferring using the difference of BLEURT scores of the two models. Oracle also tends to defer longer outputs, but the preference is moderate as compared to `Chow-Sum`. **Right**: Corresponding deferral curves.

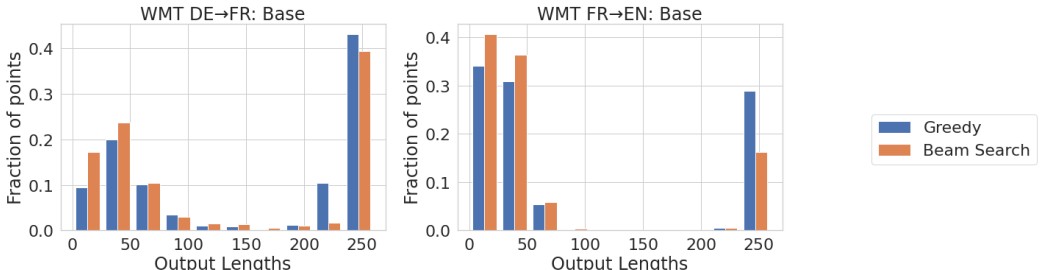

Figure 19: Comparison of Output lengths on WMT DE → FR and WMT FR → EN for a FLAN-T5 Base model with beam search decoding vs. greedy decoding. We observe that greedy decoding does seem to have overall shorter output lengths as compared to greedy decoding but still puts a considerable weight on longer outputs.

E.8   PERFORMANCE OF METHODS IN PREDICTING GOLDEN DEFERRAL LABEL

| Dataset | Chow-Sum | Chow-Average | Chow-Quantile-Best | Post-Hoc-Quantile | Post-Hoc-Embed-1 | Post-Hoc-Embed-1+2 |
|---|---|---|---|---|---|---|
| MNLI | 0.46 | 0.53 | 0.73 | 0.86 | 0.92 | 0.95 |
| TriviaQA | 0.31 | 0.53 | 0.52 | 0.68 | 0.70 | 0.71 |
| WMT DE → FR | 0.51 | 0.56 | 0.62 | 0.62 | 0.63 | 0.62 |

Table 4: The table shows the AUC-ROC score for predicting the golden binary deferring labels, which is 1 if model 2 (FLAN-T5 Large) is better than model 1 (FLAN-T5 Base), and 0 otherwise. We see that `Chow-Sum` and `Chow-Average` tend to be poorly predictive of this golden label. The recommended `Post-Hoc` deferral approaches yield significant gains in the AUC-ROC.

# F    How is $\Phi(\cdot)$ computed?

Here, we give an example of how $\Phi(\boldsymbol{x})$ is computed for each of the Post-Hoc methods.

For concreteness, suppose that the example $\boldsymbol{x}$ outputs a prediction which has 5 output tokens. Let the output log probability associated with each output token be $p_i(\boldsymbol{x})$ where $i \in \{0, 1, 2, 3, 4\}$. Then, the Chow-Sum score is $s_{\text{sum}}(\boldsymbol{x}) = \sum_i p_i(\boldsymbol{x})$ and the Chow-Average score is $s_{\text{avg}}(\boldsymbol{x}) = \frac{1}{5} \cdot \sum_i p_i(\boldsymbol{x})$. Let the various quantiles be $s_{\text{quant}}(\boldsymbol{x}, \alpha)$ where $\alpha \in [0, 0.01, 0.02, ...0.1, 0.2, 0.3, \ldots, 1.0]$. For concreteness, we take 20 quantiles.

Let the decoder embedding from model 1 averaged over all output tokens be denoted as $\text{Emb}_{\text{dec,final,avg,m1}}(\boldsymbol{x}) \in \mathbb{R}^{n_1}$. Let the decoder embedding from model 2's first layer for the first output token be denoted as $\text{Emb}_{\text{dec,int,first,m2}}(\boldsymbol{x}) \in \mathbb{R}^{n_2}$.

**Post-Hoc-Quantile** For the method Post-Hoc-Quantile,

$$\Phi(\boldsymbol{x}) = [s_{\text{sum}}(\boldsymbol{x}), s_{\text{avg}}(\boldsymbol{x}), s_{\text{quant}}(\boldsymbol{x}, \alpha_i)]$$

where we concatenate the quantities, Chow-Sum and Chow-Average scores which makes this feature 22 dimensional.

**Post-Hoc-Embed-1** For the method Post-Hoc-Embed-1,

$$\Phi(\boldsymbol{x}) = [s_{\text{sum}}(\boldsymbol{x}), s_{\text{avg}}(\boldsymbol{x}), s_{\text{quant}}(x, \alpha_i), \text{Emb}_{\text{dec,final,avg,m1}}(\boldsymbol{x})]$$

where we concatenate the quantities, Chow-Sum, Chow-Average scores and model 1's decoder embeddings which makes this feature $22 + n_1$ dimensional.

**Post-Hoc-Embed-1+2** For the method Post-Hoc-Embed-1+2,

$$\Phi(\boldsymbol{x}) = [s_{\text{sum}}(\boldsymbol{x}), s_{\text{avg}}(\boldsymbol{x}), s_{\text{quant}}(\boldsymbol{x}, \alpha_i), \text{Emb}_{\text{dec,final,avg,m1}}(\boldsymbol{x}), \text{Emb}_{\text{dec,int,first,m2}}(\boldsymbol{x})]$$

where we concatenate the quantities, Chow-Sum, Chow-Average scores, model 1's decoder embeddings and model 2's intermediate decoder embeddings which makes this feature $22 + n_1 + n_2$ dimensional.

