# OpenReview forum: "Language Model Cascades: Token-Level Uncertainty And Beyond"
_ICLR.cc/2024/Conference — ICLR 2024 poster_

### Official Review · Reviewer_Qrz5 · 2023-11-01

**Soundness:** 3 good
**Presentation:** 3 good
**Contribution:** 3 good
**Rating:** 6
**Confidence:** 4

**Summary:**

The authors aim to apply simple model cascades to structured output problems of varying length: try to first use a small model, then fall back to a larger model if it appears the small model is insufficiently confident. This task has been used to good effect on problems with simpler output spaces, such as multi-class prediction. In those settings, the log prob of the prediction can be used as a proxy for confidence; low probabilities from a small model trigger inference from a large model. However, in language generation tasks, the length can vary broadly, hence the log probabilities can vary broadly as well. The direct analog of log probabilities would be to sum the log probs of each prediction, but this has undesirable scale issues based on sequence length.

The authors propose both simple confidence estimate techniques (average, percentiles) and complex methods (learned functions of percentiles, embeddings) that lead to substantial improvements. In some settings the cascade performs better than either model alone, suggesting that the model is attaining some kind of ensemble effect.

**Strengths:**

The authors present an accessible introduction to cascades as well as the challenges of application to language generation tasks. The methods they propose are straightforward and easy to implement, and seem to work well.

The authors evaluate several different tasks, using both simple and complex models, and present reasonable gains.

The post-hoc methods provide some interesting insights.

**Weaknesses:**

The authors only work with a single base model: FLAN-T5. It's not clear how well these results generalize.

There are other methods of confidence estimation beyond logprobs (see, e.g. https://direct.mit.edu/tacl/article/doi/10.1162/tacl_a_00598/117737/Calibrated-Interpretation-Confidence-Estimation-in) -- would like to see more analysis here.

**Questions:**

In the "Intermediate embeddings" approach, only decoder representations are used. However, wouldn't it potentially be useful to characterize aspects of the input? I could see that some inputs might be more reasonable for a smaller model; others might have complexities that are more suited to a larger model. Even input length could potentially be useful. Do you have empirical experimentation here?

---

> ### Author Response · Authors · 2023-11-21
> **Response to Reviewer Qrz5**
>
> Thanks for the positive feedback and thoughtful comments.
>
> > The authors only work with a single base model: FLAN-T5. It's not clear how well these results generalize.
>
> Thanks for the suggestion. Our focus on FLAN-T5 models was motivated by their versatile performance on a range of downstream tasks (owing to instruction tuning), as well as them offering models of different sizes.
>
> > There are other methods of confidence estimation beyond logprobs (see, e.g. https://direct.mit.edu/tacl/article/doi/10.1162/tacl_a_00598/117737/Calibrated-Interpretation-Confidence-Estimation-in) -- would like to see more analysis here.
>
> Thanks for this reference. From our reading, the paper considers *min aggregation* versus *mean aggregation* for confidence measure (Section 4.4). The paper finds that *min aggregation* leads to better calibration which we have already included in this work.
>
> We note also that Appendix D covers some other approaches to estimating uncertainty. While some of these may be computationally prohibitive for a cascade setting, it would be of interest to see if others offer value over the information in softmax probabilities. This could be a good direction for future work (as noted in Section 5).
>
> > In the "Intermediate embeddings" approach, only decoder representations are used. However, wouldn't it potentially be useful to characterize aspects of the input? I could see that some inputs might be more reasonable for a smaller model; others might have complexities that are more suited to a larger model. Even input length could potentially be useful. Do you have empirical experimentation here?
>
> Yes, we had experimented with both output only, as well as input and output combined representations. But, adding input representations did not seem to help. We have added results for 3 datasets with input representations also in Appendix F.3.

---

> ### Author Response · Authors · 2023-11-22
>
> We really appreciate your time and effort in giving detailed comments and feedback. We have updated our paper based on this and have answered your questions above. Please let us know if you have any further questions or comments.

---

### Official Review · Reviewer_MA2R · 2023-11-03

**Soundness:** 3 good
**Presentation:** 3 good
**Contribution:** 3 good
**Rating:** 6
**Confidence:** 3

**Summary:**

The paper focuses on the model cascade for generation tasks. It notices a crucial difference between cascading for classification tasks and cascading for generation task and points out that the natural extension of predicted class uncertainty to generative tasks, predicted sequence uncertainty, is biased by sequence length, leading to sub-optimal deferral decisions.
It then designs a deferral rule to obtain the score/confidence of the small LM and decides when to defer an input to the larger model.

**Strengths:**

1.	It demonstrates that simple sequence-level LM confidence measures for deferral can lead to sub-optimal cost-quality tradeoffs due to length bias.
2.	The paper proposes a simple yet effective method employing the quantile of the log-likelihood to design a deferral rule.
3.	The Proposal of a post-hoc deferral rule trained on quantile features and the input embeddings of both the small LM and the large LM. The extensive experiments on FLAN-T5 verify the efficacy of the method.

**Weaknesses:**

1.	Compared with simple averaging the log probability, the major advantage of quantile is that it reflects more about the overall log probability distribution of the sequence and is more robust to outliers. To highlight the motivation of the proposal, more evidence for the existence of the outlier is expected and the showcase in Figure 1 is not sufficient.
2.	In Figure 2 and Figure 3, it seems that the best generation performance is obtained in the middle of the curve, other than the endpoint where all examples are deferred. Does this mean that sometimes smaller LM outperform larger ones?
3.	The author claims that Chow-Sum is overly biased towards deferring longer predictions. However, from Figure3(a) we can obverse that when the output length (y-axis) is between 150 words to 250 words, the score of the oracle deferring strategy is smaller than the Chow-Sum, which says the opposite: the chow-sum isn’t biased towards deferring longer predictions when compared with the oracle.
4.	As another line of work, speculative decoding also aims at the trade-off balance between efficiency and performance and I think the authors should discuss or compare the difference between these two lines of work.

**Questions:**

1.	What is the performance of Post-Hoc-Embed-1?
2.	How is the $\Phi(x)$ computed in detail?
3.	What is the performance of the post-hoc-quantile when predicting the golden deferring label?
4.	In figure2, figure3 and figure5, all figures use the deferring rate as the x-axis. I am curious about whether we could use the inference time cost as the x-axis.

---

> ### Author Response · Authors · 2023-11-21
> **Response to Reviewer MA2R**
>
> Thanks for the detailed feedback and insightful comments.
>
> > Compared with simple averaging the log probability, the major advantage of quantile is that it reflects more about the overall log probability distribution of the sequence and is more robust to outliers. To highlight the motivation of the proposal, more evidence for the existence of the outlier is expected and the showcase in Figure 1 is not sufficient.
>
> We wish to emphasize that our goal in leveraging probability quantiles is not only to achieve robustness to outliers: it is to obtain a richer view of the distribution than a single summary statistic (like the sum or mean). For example, while the sum of log-probabilities may be low, if even a couple of tokens have high uncertainty, that could indicate a prediction is uncertain (per Figure 1).
>
> **Our experimental results confirm this goal is worthwhile**: by learning a post-hoc deferral rule on top of quantiles, one can significantly improve the cost-quality tradeoff compared to simply using a single summary statistic (see Table 1).
>
> > In Figure 2 and Figure 3, it seems that the best generation performance is obtained in the middle of the curve, other than the endpoint where all examples are deferred. Does this mean that sometimes smaller LM outperform larger ones?
>
> Thanks for raising this: indeed, this indicates that **the small model may be superior to the larger one on certain examples**. For example, on TriviaQA, we find that on 1.1% of test examples, the FLAN-T5 Base model is correct while the FLAN-T5-Large model is incorrect. This “non-monotonicity” of the performance of models of different sizes has been observed previously, e.g.,
>
> Narayan et al., Predicting on the Edge: Identifying Where a Larger Model Does Better. 2022.
> Kim et al., Speculative Decoding with Big Little Decoder, 2023,
>
> > The author claims that Chow-Sum is overly biased towards deferring longer predictions. However, from Figure3(a) we can observe that when the output length (y-axis) is between 150 words to 250 words, the score of the oracle deferring strategy is smaller than the Chow-Sum, which says the opposite: the chow-sum isn’t biased towards deferring longer predictions when compared with the oracle.
>
> **We believe there is a misunderstanding in the interpretation of Figure 3(a).** The x-axis denotes the score quantile for different methods. At a given score quantile (say, x=0.2), we consider the behavior of each method when the corresponding fraction of lowest scoring samples are deferred (e.g., samples with lowest 20% scores are deferred). The y-axis represents the average output length of the deferred samples.
>
> At x=0.2, the Chow-Sum method has a y value of 250, whereas the Oracle method has a y value of around 125. So, this means that the first 20% of samples which Chow-Sum defers have an average length of 250, whereas the Oracle method’s deferred samples have an average length of 125. Thus, Chow-Sum prefers to defer longer predictions.
>
> To further illustrate this, we have also added more plots in Appendix F.1 which shows the histogram and CDFs of lengths of outputs deferred by each method at different deferral rates.  Each plot considers the samples which are deferred by each method at the mentioned deferral rate. Figure 11 and 12 shows that the Oracle method has a higher fraction of points deferred with lower prediction lengths as compared to the Chow-Sum method at different deferral rates. **These plots conclusively provide evidence that Chow-Sum is overly biased towards deferring longer predictions as compared to the Oracle method.**

---

> ### Author Response · Authors · 2023-11-21
> **Response to Reviewer MA2R**
>
> > As another line of work, speculative decoding also aims at the trade-off balance between efficiency and performance and I think the authors should discuss or compare the difference between these two lines of work.
>
> Thanks for raising this interesting point. Indeed, the recent line of work on speculative decoding is conceptually related to cascading: both approaches involve orchestrating between a small and large model to improve inference efficiency.
>
> However, they have slightly different use-cases: speculative decoding critically assumes that it is feasible to use the large “verifier” model to score predictions from the small “drafter” model. Note that such scoring needs to be done *for every test example*. However, this may not always be feasible: e.g., consider a setting where the large model has >100B parameters, which could result in a high latency even when scoring sequences. By contrast, cascading can still be applicable in such cases, since the large model needs only be invoked on samples where the deferral mechanism predicts to forward.
>
> On the flip side, an appealing characteristic of speculative decoding is that it is provably quality-neutral. On the other hand, cascading offers a *tradeoff* between quality and cost. (Typically, it is empirically observed that the cascade can also be quality neutral at moderate deferral rate to the large model.) It is of considerable interest to study mechanisms that combine the strengths of cascading and speculative decoding; however, this would be worthy of a separate paper in itself.
>
> >  What is the performance of Post-Hoc-Embed-1?
>
> Thanks for pointing this out. We did not include this in the main paper for the sake of clarity and readability. Nonetheless, here is the performance comparison, summarized via the AUC values for each dataset, and percentage improvement over random.
> | Dataset | Post-Hoc-Embed-1 AUC-DF (% improvement over random) |
> |----------|-------------------|
> |anli-r1 |0.546 (+ 8.80)|
> |anli-r2 | 0.450 (+ 2.21) |
> |anli-r3 |0.426 (+ 6.07)|
> | boolq | 0.840 (+ 2.96)|
> | imdb |0.964 (+ 1.23) |
> |lambada |0.693 (+ 2.07) |
> |mnli |0.719 (+ 11.82) |
> |piqa |0.718 (+ 2.17)|
> |squad|0.408 (+ 1.65)|
> |triviaqa |0.099 (+ 17.65)|
> |tydiqa-fi|0.333 (+ 4.47)|
> |tydiqa-id|0.242 (+ 4.62)|
> |tydiqa-sw|0.162 (+ 5.47)|
> |winogrande|0.644 (+ 1.02)|
> |wmt-de-fr| 0.404 (+ 4.23)|
> |wmt-en-fr| 0.444 (+ 3.40)|
> |wmt-fr-en| 0.618 (+ 2.99)|
>
> We see that Post-Hoc-Embed-1 is able to improve upon Post-Hoc-Quantile and Chow-Quantile methods slightly, but is slightly inferior compared to the Post-Hoc-Embed-1+2 method. This intuitively makes sense as this has more information compared to the Post-Hoc-Quantile method but still does not include any information about model 2. We have also added this text in Section 4.3 in the draft.

---

> > ### Author Response · Authors · 2023-11-21
> > **Response to Reviewer MA2R**
> >
> > > How is the $\Phi(x)$ computed in detail?
> >
> > We will clarify how $\Phi(x)$ is computed for each of the Post-Hoc methods with the help of an example.
> >
> > Let us say that the example $x$ outputs a prediction which has 5 output tokens. Let the output log probability associated with each output token be $p_i(x)$, where $0 <= i < 4$. Then, the Chow-Sum score is $s_{\rm sum}(x) = \sum_i p_i(x)$ and the Chow-Average score is $s_{\rm avg}(x) =\frac{1}{5} \sum_i p_i(x)$. Let the various quantiles be $s_{\rm quant}(x,\alpha_i)$ where $\alpha_i$ is in [0, 0.01, 0.02,... 0.1, 0.2, 0.3, …, 1.0]. We take 20 quantiles.
> >
> > Let the decoder embedding from model 1’s final layer (averaged over all output tokens) be denoted as ${\rm Emb}_{\rm dec,final,avg,m1}(x) \in \mathbb{R}^{n_1}$.
> >
> > Let the decoder embedding from model 2’s first layer (for the first output token) be denoted as ${\rm Emb}_{\rm dec,int,first,m2}(x) \in \mathbb{R}^{n_2}$.
> >
> > **Post-Hoc-Quantile**
> >
> > For the method Post-Hoc-Quantile, we concatenate the quantiles, Chow-Sum and Chow-Average scores:
> >
> > $$ \Phi(x) = [ s_{\rm sum}(x), s_{\rm avg}(x), \{ s_{\rm quant}(x,\alpha_i) \} ] \in \mathbb{R}^{22} $$
> >
> > **Post-Hoc-Embed-1**
> >
> > For the method Post-Hoc-Embed-1, we concatenate the quantiles, Chow-Sum, Chow-Average scores and model 1’s decoder embeddings:
> >
> > $$ \Phi(x) = [ s_{\rm sum}(x), s_{\rm avg}(x), \{ s_{\rm quant}(x,\alpha_i) \}, {\rm Emb}_{\rm dec,final,avg,m1}(x) ] \in \mathbb{R}^{22 + n_1} $$
> >
> > **Post-Hoc-Embed-1+2**
> >
> > For the method Post-Hoc-Embed-1+2, we concatenate the quantiles, Chow-Sum, Chow-Average scores, model 1’s decoder embeddings and model 2’s intermediate decoder embeddings:
> >
> > $$ \Phi(x) = [ s_{\rm sum}(x), s_{\rm avg}(x), \{ s_{\rm quant}(x,\alpha_i) \}, {\rm Emb_{\rm dec,final,avg,m1}}(x), {\rm Emb_{\rm dec,int,first,m2}}(x) ] \in \mathbb{R}^{22 + n_1 + n_2} $$
> >
> > We have also added this example in Appendix F.5.
> >
> > > What is the performance of the post-hoc-quantile when predicting the golden deferring label?
> >
> > Thanks for the interesting question. The final performance of a cascade is typically measured by the accuracy of the interleaved predictions from the small and large model. Nonetheless, the reviewer’s suggestion is indeed a useful diagnostic.
> >
> >
> > To that end, the table below shows the AUC-ROC score for predicting the golden binary deferring labels, which is 1 if model 2 is better than model 1, and 0 otherwise. We see that Chow-Sum and Chow-Average tend to be poorly predictive of this golden label. The recommended Post-Hoc deferral approaches yield significant gains in the AUC-ROC. We also added the table in Appendix F.6.
> >
> > | Method |MNLI | TriviaQA | WMT DE -> FR|
> > |-|-|-|-|
> > |Chow-Sum|0.46|0.31|0.51|
> > |Chow-Average|0.53|0.53|0.56|
> > |Chow-Quantile-Best|0.73|0.52|0.62|
> > |Post-Hoc-Quantile|0.86|0.68|0.62|
> > |Post-Hoc-Embed-1|0.92|0.70|0.63|
> > |Post-Hoc-Embed-1+2|0.95|0.71|0.62|
> >
> >
> >
> >
> > > In figure2, figure3 and figure5, all figures use the deferring rate as the x-axis. I am curious about whether we could use the inference time cost as the x-axis.
> >
> > Thanks for the suggestion. Note that the two are linearly related: for a deferral rate of $d%%$, the inference cost is $c_{\rm S} \cdot \frac{100 - d}{100} + (c_{\rm S} + c_{\rm L}) \cdot \frac{d}{100}$, where $c_{\rm S}, c_{\rm L}$ are the costs of the small and large model respectively.
> > We have also added a plot in Appendix F.2 where we plot the relative latency on the x-axis. In our experiments, we estimate the base model latency to be 36% that of the large model. (Please note that the latency estimate may vary depending on the dataset, because the number of output tokens varies. We estimate the latency based on 500 random examples on XSum dataset; this is intended to serve as a rough estimate of the relative speedup from cascading.)

---

> ### Author Response · Authors · 2023-11-22
>
> We really appreciate your time and effort in giving detailed comments and feedback. We have updated our paper based on this and have answered your questions above. Please let us know if you have any further questions or comments.

---

### Official Review · Reviewer_WdbA · 2023-11-08

**Soundness:** 3 good
**Presentation:** 3 good
**Contribution:** 2 fair
**Rating:** 8
**Confidence:** 3

**Summary:**

The authors propose a novel method for uncertainty estimation in large language models (LLM). The method is based on LLM cascades, where a large model predicts difficult instances and a second small model predicts easy instances. In addition, the model uses the Chow-sum and Chow-average as a rule for assigning confidence (uncertainty) to token-level outputs from the LLM cascade.  The main contributions are: i) method for token-level uncertainty estimates, and ii) application of different natural language processing (NLP) tasks. The method shows that the confidence estimates based on the output probabilities from LLMs are biased.

**Strengths:**

- A principled method for uncertainty estimation in LLM (i.e. FLAN-T5).
- Clear description of background knowledge and related work needed to understand the proposed method.
- The authors perform a  comprehensive comparison of the proposed method with different NLP tasks.

**Weaknesses:**

- Motivation for the lack of comparison with other uncertainty estimation methods.
- A possible extra contribution can be the use or discussion of the method for NLP tasks under out-of-distribution (OOD) or domain adaptation.

**Questions:**

Please address the following questions during the rebuttal:

- Please elaborate on the relation/difference of the Chow estimates with proper scoring rules (e.g. NLL, Brier score).
- Could the proposed estimates be directly compared/evaluated to estimates from deep ensembles instead of a cascade or even jointly? (Lakshminarayanan, Balaji et al. “Simple and Scalable Predictive Uncertainty Estimation using Deep Ensembles.” Neural Information Processing Systems (2016).)
- For the machine translation evaluation:

 Is the output generated by beam search? Please speculate on the effect of the hyperparameters used on the generation, do they have an effect on the output length?

 Please speculate for the use of the proposed method for robustness to OOD in MT. different domains can be used to evaluate a change in distribution, is the uncertainty estimate robust to such change?

- Please elaborate on the use of output probabilities from the LLMs as uncertainty estimates compared to other methods (e.g. deep enembles, MC dropout)? (e.g. Baan, Joris et al. “Uncertainty in Natural Language Generation: From Theory to Applications.” ArXiv abs/2307.15703 (2023): n. pag.)

**Details Of Ethics Concerns:**

I have no concerns.

---

> ### Author Response · Authors · 2023-11-21
> **Response to Reviewer WdbA**
>
> Thanks for the detailed feedback and encouraging comments!
>
> > Please elaborate on the relation/difference of the Chow estimates with proper scoring rules (e.g. NLL, Brier score).
>
> This is a good question. A proper loss (or negative proper scoring rule) is typically defined for a predicted probability *and* a ground truth label; e.g., we may compute the log-loss $-log \hat{p}(y \mid x)$ for a predicted distribution $\hat{p}( \cdot \mid x )$ and label $y$.
>
> In our setting, we only have the predicted distribution at inference time; the ground truth label $y$ is unknown, and so cannot be used to compute a confidence measure for deferral. With Chow’s rule, we compute the maximum value of $\hat{p}( y’ \mid x )$ over *all possible* labels $y’$.
>
> > Could the proposed estimates be directly compared/evaluated to estimates from deep ensembles instead of a cascade or even jointly? (Lakshminarayanan, Balaji et al. “Simple and Scalable Predictive Uncertainty Estimation using Deep Ensembles.” Neural Information Processing Systems (2016).)... Please elaborate on the use of output probabilities from the LLMs as uncertainty estimates compared to other methods (e.g. deep enembles, MC dropout)? (e.g. Baan, Joris et al. “Uncertainty in Natural Language Generation: From Theory to Applications.” ArXiv abs/2307.15703 (2023): n. pag.)
>
> Thanks for the interesting suggestion. First, we note that cascades and ensembles solve slightly different problems. Cascading is often used as a way of improving the prediction *efficiency*, by selectively leveraging *one* model from a given family. Ensembling is often used as a way of improving the prediction *quality*, by suitably leveraging *multiple* models from a given family.
>
> Deep ensembles are generally understood to provide more reliable uncertainty estimates than a single model. However, this comes at the expense of a higher compute cost than invoking a single model. For cascading, efficiency is paramount: such a higher cost would defeat any gains obtained by the improved uncertainty estimates. A similar concern would arise in approaches based on averaging multiple models using dropout.
>
> One can in fact combine cascading and ensembling, per Wang et al., “Wisdom of Committees: An Overlooked Approach To Faster and More Accurate Models”. However, such approaches have not (to our knowledge) been explored in the language model domain. We agree this could be extremely interesting for future work!
>
> > Motivation for the lack of comparison with other uncertainty estimation methods.
>
> As discussed above, many uncertainty estimation procedures involve additional computation (e.g., multiple inferences with a single model in dropout-based approaches and single inference with multiple models in ensemble-based approaches) compared to simply using softmax probability outputs from a single network. Such approaches are less appealing for use in cascades, where the primary goal is to improve efficiency.
> We have also updated the text in Appendix D to include this discussion.
>
> > For the machine translation evaluation: Is the output generated by beam search?
>
> The outputs were generated by greedy decoding. To verify that the results are not an artifact of this decoding strategy, we have also added results for beam search in Appendix F.4 in Figures 15, 16 and 17. The output lengths are on average lower in beam search as compared to greedy decoding. We observe that the same findings still hold in this case.
>
> > Please speculate on the effect of the hyperparameters used on the generation, do they have an effect on the output length?
>
> Thanks for raising this important point. We added a plot (Figure 18) in Appendix F.4 which compares the distribution of output lengths using greedy decoding versus beam search (beam size 10). As our intuition suggests (and also pointed out by Reviewer 7G2G), we see that outputs generated by beam search have lower lengths on average as compared to the outputs generated by greedy decoding because they are less prone to get stuck in repetitions. But, as the figure suggests, there is still a considerable probability mass on longer lengths and hence, they are still likely to generate longer outputs with repetitive text. Thus, as the rest of the figures in Appendix F.4 suggest, our insights and methods still generalize to outputs generated using beam search.

---

> > ### Comment · Reviewer_WdbA · 2023-11-21
> >
> > Thank you for addressing my questions. I have no further comments.

---

> > > ### Author Response · Authors · 2023-11-21
> > > **Thank you**
> > >
> > > Thank you for the update. We really appreciate your time and effort.

---

> ### Author Response · Authors · 2023-11-21
> **Response to Reviewer WdbA**
>
> > Please speculate for the use of the proposed method for robustness to OOD in MT. different domains can be used to evaluate a change in distribution, is the uncertainty estimate robust to such change?
>
> Thanks for the interesting suggestion. We note that the distinction between OOD and ID data is not immediately clear in our setting of instruction tuned LMs. Specifically, we are operating in a setting where there is a single LM that is first pre-trained, and then fine-tuned on a given instruction-tuning set (the FLAN mixture for FLAN-T5). The latter implicitly contains multiple tasks, and aims to help the model generalize to new tasks.
> One narrow definition of “OOD” in this setting could be based on whether or not a task is included in the instruction-tuning mixture. For example, the TyDiQA dataset is not included in the FLAN mixture. One might thus consider TyDiQA as, in a narrow sense, “OOD” for FLAN-T5 models. We have included results for TyDiQA in our paper. On one of the three languages for TyDiQA, our method outperforms other methods.
> For OOD in machine translation, if the reviewer has a specific task in mind, we would be happy to consider experimenting with it.

---

### Official Review · Reviewer_7G2G · 2023-11-08

**Soundness:** 3 good
**Presentation:** 4 excellent
**Contribution:** 3 good
**Rating:** 8
**Confidence:** 4

**Summary:**

This paper is about learning deferral rules for LM cascades -- essentially, how can you predict when to rely on a small model's output and when should you fall back to a large, more expensive model? They propose a connection to the classical problem of classification with rejection, in which a model can choose to reject classifying an instance. The optimal strategy in this case is to reject whenever the model's confidence fails to pass a threshold derived from the cost of rejection. It is easy to see how LM cascades fit into this framework: deferring to the larger model is equivalent to rejecting the output of the smaller model.

However, it is not trivial to generalize rejection from classification to generation. In classification the model predicts only one label along with an easy-to-interpret probability score (e.g. from softmax), whereas generation entails producing variable-length *sequences* of tokens. Although each of these tokens has an associated per-token probability, aggregating them is a challenge: simply summing the logprobs causes short sequences to be rejected (this is mathematically necessary, as it is equivalent to multiplying numbers less than one), while averaging them causes *long* sequences to be rejected (this observation is interesting, and surprising to me).

Noting the weaknesses of these two baselines, the authors' main contribution is to introduce alternative techniques to score the model output. The main technique, which they call Chow-Quantile, is based on sorting the per-token logprobs for an instance and then picking the value at some alpha-quantile of this, where alpha is a hyperparameter. For example, picking the 0-quantile is equivalent to scoring sequences based on the *lowest* per-token prob. They also propose various post-hoc techniques that allow classifiers to be trained on top of these quantiles.

They exhibit experiments applying their various deferral techniques to a variety of tasks, including MT and QA (and also surprisingly MNLI, which seems inapplicable because it classification, not generation). These results seem to show an advantage for using their techniques over the baselines across many deferral levels.

**Strengths:**

This paper presents a simple approach that seems to be very effective. The connection to rejection in classifiers is intuitive but had not occurred to. The paper is easy to follow. The clarity of presentation convinces me that it would be easy for me to try the approach myself, either for its own utility or as a replication study. I admire the accessibility of this work.

**Weaknesses:**

Although the paper as a whole is very clear, there are places where the experiments lack specifics (see the questions section). Additionally, there are places where the experimental set-up seems to be suboptimal: greedy decoding was used, but this is more prone to hallucination than beam search. Some of the conclusions of the experiments might be artifacts of these hallucinations. I can think specifically of these two:

1) there was a negative correlation between translation quality and sequence length; was this because many of the long sequences were hallucinations?

2) the Chow-average model favored longer sequences, which there is no intuitive reason for. Could it be because hallucinations often get trapped in loops where the same short phrases get repeated with high probability? This would drive up the average.

**Questions:**

--Which WMT set was used? It needs to be identified and cited.

--I don't understand how the MNLI experiments work. It is noted in Section 3.4 that MNLI is a multi-class classification problem, but the techniques proposed in this paper are not for classification.

---

> ### Author Response · Authors · 2023-11-21
> **Response to Reviewer 7G2G**
>
> Thanks for the detailed feedback and encouraging comments!
>
> > Additionally, there are places where the experimental set-up seems to be suboptimal: greedy decoding was used, but this is more prone to hallucination than beam search.
>
> **We have added cascade plots with beam search** for three datasets (TriviaQA, WMT DE->FR and WMT FR->EN; see Appendix F.4, Figures 15, 16), and a length analysis plot for WMT FR->EN (Appendix F.4, Figure 17). We used a beam size of 10 for this experiment. The output lengths are on average lower in beam search as compared to greedy decoding (as shown in Figure 18 in Appendix F.4 as suggested by reviewer WdbA) which gives more support to the reviewer’s intuition that beam search is less likely to get caught in repetitive hallucinations as compared to greedy decoding but still gets affected by this problem. Thus, we observed similar conclusions to our main results. Thus, our results also extend to decoding with beam search, and are not purely an artifact of greedy decoding.
>
> > the Chow-average model favored longer sequences, which there is no intuitive reason for. Could it be because hallucinations often get trapped in loops where the same short phrases get repeated with high probability? This would drive up the average.
>
> We believe the reviewer has the correct intuition. As noted, Chow-average preferred keeping **longer** sequences and deferring **shorter** sequences (see (0, 0.2) quantile score range of Figure 3a). Longer sequences got a higher Chow-average score on average for two reasons: (a) the later tokens tend to have higher probability as per Figure 3c, and (b) as the reviewer suggests, there are loops where the same short phrase gets repeated. That is why, after normalization by the sequence length, Chow-average prefers these longer sequences.
>
> > there was a negative correlation between translation quality and sequence length; was this because many of the long sequences were hallucinations?
>
> Per discussion in Section 3.5 (see also Figure 4), longer predictions tend to have repetitions, which results in lower translation quality.
>
> > --Which WMT set was used? It needs to be identified and cited.
>
> Thanks for pointing this out. We have used WMT 14 FR-> EN, WMT 14 EN-> FR and WMT 19 DE-> FR. We also updated the draft to include these citations in Section 3.4.
>
> > --I don't understand how the MNLI experiments work. It is noted in Section 3.4 that MNLI is a multi-class classification problem, but the techniques proposed in this paper are not for classification.
>
> Sorry for the confusion. Following the T5 and FLAN-T5 papers, we treat *all* problems as finding a text to text mapping. So for MNLI, we encode the classes as strings, namely, “entailment”, “neutral”, “contradiction”. We take the model’s output text and perform a string comparison to the label. We have updated the draft to clarify this.

---

> > ### Comment · Reviewer_7G2G · 2023-11-21
> >
> > Thank you for the informative response. I have adjusted my score.

---

> > > ### Author Response · Authors · 2023-11-21
> > > **Thanks**
> > >
> > > Thank you for the update. We really appreciate your time and effort.

---

### Author Response · Authors · 2023-11-21
**Common Response**

Common Response

We would like to thank all the reviewers for their detailed feedback and thoughtful comments which have been helpful in improving our draft. Reviewers consistently appreciated the simplicity and effectiveness of our approach (7G2G, MA2R, Qrz5), the clarity of presentation (7G2G, WdbA, Qrz5) and comprehensiveness of our experiments and gains (7G2G, WdbA, MA2R, Qrz5).

We have updated the paper taking into account the reviewer feedback and comments. **We have highlighted the changes in the main paper (in blue) and added additional results in Appendix F.**

**Additional results with beam search:** Reviewers 7G2G and WdbA commented on the importance of greedy decoding for our results. In response to this, we have added additional results in Appendix F.4 with beam search with similar conclusions as those with greedy decoding. These actually showcase that our results and analyses are not tied to greedy decoding.

**Additional results with encoder embeddings:** As suggested by reviewer Qrz5, we have added results in Appendix F.3. for additional baselines which use encoder embeddings in the post hoc deferral approaches. These comparisons show that including encoder embeddings along with decoder embeddings do not provide any additional gains.

**Additional visualizations of results:** We have added new visualizations of our length analysis results in Appendix F.1 to help the readers gain a different perspective: it allows them to clearly see that Chow-sum prefers to defer longer predictions as compared to the oracle method. Thanks a lot to reviewer MA2R for this suggestion.

**Cascade curves with latency on x-axis:** As suggested by reviewer MA2R, we have added cascade plots with relative inference latency on the x-axis in Appendix F.2. We can clearly see that Chow-Quantile method can significantly improve upon the inference latency at a fixed accuracy level.

---

### Meta-Review · Area_Chair_7rA5 · 2023-12-14

**Metareview:**

This paper proposes a cascade LM approach, where “easy” instances are handled by a small model, while a few “hard” instances are deferred to the large model. The paper presents a systematic study of deferral rules for LM cascades. The weaknesses raised by the reviewers have mostly been addressed at the rebuttal stage. Overall, all reviewers feel positively about this paper and so do I. I recommend acceptance.

**Justification For Why Not Higher Score:**

Does not seem an extremely innovative idea.

**Justification For Why Not Lower Score:**

Already mentioned in the review.

---

### Decision · Program_Chairs · 2024-01-16

Accept (poster)